# Towards space-borne monitoring of localized $CO_2$ emissions: an instrument concept and first performance assessment

Johan Strandgren[1], David Krutz[2], Jonas Wilzewski[1,3], Carsten Paproth[2], Ilse Sebastian[2], Kevin R. Gurney[4], Jianming Liang[5], Anke Roiger[1], and André Butz[6]

[1]Deutsches Zentrum für Luft- und Raumfahrt, Institut für Physik der Atmosphäre, Oberpfaffenhofen, Germany
[2]Deutsches Zentrum für Luft- und Raumfahrt, Institut für Optische Sensorsysteme, Berlin-Adlershof, Germany
[3]Meteorological Institute Munich, Ludwig-Maximilians-Universität, Munich, Germany
[4]School of Informatics, Computing and Cyber Systems, Northern Arizona University, Flagstaff, AZ, USA
[5]School of Life Sciences, Arizona State University, Tempe, AZ, USA
[6]Institut für Umweltphysik, Universität Heidelberg, Heidelberg, Germany

*Correspondence to:* Johan Strandgren (johan.strandgren@dlr.de)

**Abstract.** The UNFCCC (United Nations Framework Convention on Climate Change) requires the nations of the world to report their carbon dioxide ($CO_2$) emissions. Independent verification of these reported emissions is a corner stone for advancing towards emission accounting and reduction measures agreed upon in the Paris agreement. In this paper, we present the concept and first performance assessment of a compact space-borne imaging spectrometer with a spatial resolution of $50 \times 50\,\mathrm{m}^2$ that

could contribute to the "monitoring, verification and reporting" (MVR) of $CO_2$ emissions worldwide. $CO_2$ emissions from medium-sized power plants ($1$–$10\,\mathrm{MtCO_2\,yr^{-1}}$), currently not targeted by other space-borne missions, represent a significant part of the global $CO_2$ emission budget. In this paper we show that the proposed instrument concept is able to resolve emission plumes from such localized sources as a first step towards corresponding $CO_2$ flux estimates.

    Through radiative transfer simulations, including a realistic instrument noise model and a global trial ensemble covering

various geophysical scenarios, it is shown that an instrument noise error of 1.1 ppm ($1\sigma$) can be achieved for the retrieval of the column-averaged dry-air mole fraction of $CO_2$ ($XCO_2$). Despite limited amount of information from a single spectral window and a relatively coarse spectral resolution, scattering by atmospheric aerosol and cirrus can be partly accounted for in the $XCO_2$ retrieval, with deviations of at most 4.0 ppm from the true abundance for two thirds of the scenes in the global trial ensemble.

We further simulate the ability of the proposed instrument concept to observe $CO_2$ plumes from single power plants in an urban area using high-resolution $CO_2$ emission and surface albedo data for the city of Indianapolis. Given the preliminary instrument design and the corresponding instrument noise error, emission plumes from point sources with an emission rate down to the order of $0.3\,\mathrm{MtCO_2\,yr^{-1}}$ can be resolved, i.e. well below the target source strength of $1\,\mathrm{MtCO_2\,yr^{-1}}$. This leaves a significant margin for additional error sources like scattering particles and complex meteorology and shows the potential for

subsequent $CO_2$ flux estimates with the proposed instrument concept.

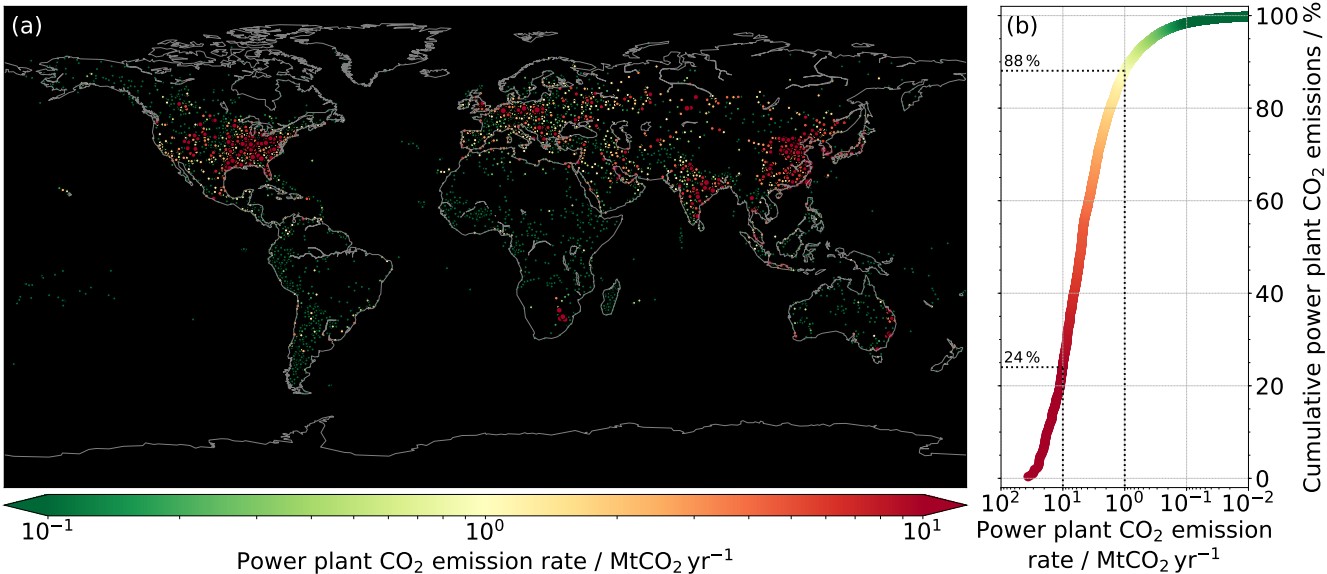

**Figure 1. (a)** Geographical distribution of reported and estimated annual $CO_2$ emissions from power plants worldwide for the year 2009, as provided by the CARMA v3.0 database. **(b)** Corresponding cumulative distribution showing the fraction of the power plant emission total ($9.9\,GtCO_2\,yr^{-1}$) that power plants with a source strength greater than $X\,MtCO_2\,yr^{-1}$ make up. This should be understood as the fraction of the power plant $CO_2$ emission total that, theoretically, can be observed by an instrument with a given sensitivity. For visualization purposes, the marker sizes in **(a)** are scaled according to the respective emission rates.

## 1 Introduction

Despite the broad consensus on the negative long-term effects of carbon dioxide ($CO_2$) emissions and the efforts reducing these emissions, the atmospheric $CO_2$ concentrations continue to rise. During the course of 2018, the average $CO_2$ concentration increased from 407 to 410 ppm at the Mauna Loa observatory, representing the fourth-highest annual growth ever recorded at that observatory (NOAA, 2019). $CO_2$ emissions from localized point sources represent a large fraction of the $CO_2$ emitted into the atmosphere. The International Energy Agency (IEA) recently reported that emissions from coal-fired power plants exceeded $10\,GtCO_2\,yr^{-1}$ for the first time in 2018, hence accounting for approx. 30 % of the global $CO_2$ emissions (IEA, 2019), mainly due to continued growth of coal use in Asia and other emerging economies. Figure 1 depicts the global distribution of reported and estimated annual $CO_2$ emissions from power plants for the year 2009, as provided by the CARMA (Carbon Monitoring for Action) v3.0 database (Wheeler and Ummel, 2008; Ummel, 2012), together with the corresponding cumulative distribution of the power plant emissions. The emission total from 16 898 individual power plants, where exact or approximate coordinates are available, adds up to $9.9\,GtCO_2\,yr^{-1}$. A large fraction of power plant emissions originates from a relatively small number of large to medium-sized power plants. The CARMA data show that 153 large power plants ($> 10\,MtCO_2\,yr^{-1}$) accounted for 24 % of the total annual power plant $CO_2$ emissions, whereas 2111 large and medium-sized power plants ($> 1\,MtCO_2\,yr^{-1}$)

accounted for as much as 88 % of the power plant $CO_2$ emission budget, clearly manifesting the significant contribution from the medium-sized power plants (1–10 $MtCO_2$ $yr^{-1}$) to the global $CO_2$ emission budget.

To advance towards emission accounting and reduction measures, agreed upon in the Paris agreement in force since 2016, independent verification of reported emissions is of high importance. To this end, space-borne instruments provide a suit-
able platform where continuous long-term measurements can potentially be combined with a near-global coverage with no geopolitical boundaries.

Most of the currently operating, planned and proposed instruments for passive $CO_2$ observations from space measure the reflected short-wave infrared (SWIR) solar radiation in several spectral windows covering the oxygen-A ($O_2$A) band near 750 nm as well as the weak and strong $CO_2$ absorption bands near 1600 and 2000 nm, respectively, e.g. GOSAT (Greenhouse
Gases Observing Satellite; Kuze et al., 2009, 2016), OCO-2 (Orbiting Carbon Observatory-2; Crisp et al., 2004, 2017), TanSat (Liu et al., 2018), GOSAT-2 (Nakajima et al., 2012), OCO-3 (Eldering et al., 2019), MicroCarb (Buil et al., 2011), GeoCarb (Moore III et al., 2018), CarbonSat (Bovensmann et al., 2010; Buchwitz et al., 2013) and G3E (Geostationary Emission Explorer for Europe; Butz et al., 2015). These instruments and instrument concepts further rely on a comparatively high spectral resolution on the order of approx. $0.05 - 0.3$ nm representing resolving powers (ratio of wavelength over the full-width half-
maximum of the instrument spectral response function) ranging from approx. 3600 for the strong $CO_2$ absorption bands near 2000 nm for CarbonSat (Buchwitz et al., 2013) up to > 20 000 for the OCO and GOSAT instruments. Such advanced instruments, like for example GOSAT and OCO-2 that have been operating since 2009 and 2014, respectively, generally target an accuracy and coverage sufficient to study the natural $CO_2$ cycle on a regional to continental scale (e.g. Guerlet et al., 2013; Maksyutov et al., 2013; Parazoo et al., 2013; Eldering et al., 2017; Chatterjee et al., 2017; Liu et al., 2017), but have also been
used to observe and quantify $CO_2$ gradients on the regional scale caused by anthropogenic $CO_2$ emissions in urban areas (Kort et al., 2012; Hakkarainen et al., 2016; Schwandner et al., 2017; Reuter et al., 2019). OCO-2 data have further been used to observe strong $CO_2$ plumes from localized natural and anthropogenic $CO_2$ sources like volcanoes and coal-fired power plants (Nassar et al., 2017; Schwandner et al., 2017; Reuter et al., 2019), demonstrating the capabilities of imaging spectrometers to monitor $CO_2$ from space. The spatial resolution of OCO-2 and similar instruments like e.g. OCO-3, TanSat and the planned
Copernicus $CO_2$ Monitoring mission CO2M (on the order of approx. 2–4 $km^2$) does, however, pose a difficulty for the routine monitoring of localized power plant $CO_2$ emissions, since the plume is usually only sampled by a handful of pixels, where $CO_2$ plume enhancements cannot be fully separated from the background, making quantitative $CO_2$ emission rate estimates difficult and vulnerable to cloud contamination and instrument noise propagating into $CO_2$ retrieval errors. For this reason CO2M will target isolated large power plants ($\gtrsim 10$ $MtCO_2$ $yr^{-1}$) and large urban agglomerations ($\gtrsim$ Berlin) (Kuhlmann et al.,
2019) and thus, a large fraction of the emission total will be missed.

To contribute to closing this gap and expanding on the future $CO_2$ monitoring from space, we here present the concept and a first performance assessment of a space-borne imaging spectrometer that could be deployed for the dedicated monitoring of localized $CO_2$ emissions. By targeting power plants with an annual emission rate down to approx. 1 $MtCO_2$ $yr^{-1}$, a substantial fraction of the $CO_2$ emissions from power plants and hence a significant part of the global man-made $CO_2$ emission budget in
total could be resolved (given a global coverage through a fleet of instruments). As shown in Fig. 1, it is of key importance to

cover also the medium-sized power plants ($1$–$10\,\mathrm{MtCO_2\,yr^{-1}}$) as they alone contributed to approx. 64 % of the $CO_2$ emissions from power plants in 2009, according to the CARMA v3.0 data. To achieve this, the proposed instrument has an envisaged spatial resolution of $50 \times 50\,\mathrm{m^2}$. With such a high spatial resolution and large amount of ground pixels per unit area, averaging of plume enhancements and background concentration fields is avoided. This leads to an enhanced contrast compared to a

coarser spatial resolution. To increase the number of collected photons and hence the signal-to-noise ratio (SNR) and relative precision of the $CO_2$ concentration retrievals, such a high spatial resolution has to be compensated for with a rather coarse spectral resolution. To further compensate for the limited spatial coverage of a single instrument, a comparatively compact and low-cost instrument design is an important aspect, as it would allow for a fleet of instruments to be deployed, increasing the spatial coverage.

Wilzewski et al. (2020) recently demonstrated that atmospheric $CO_2$ concentrations can be retrieved with an accuracy $< 1\,\%$ using such a comparatively simple spectral set-up with one single spectral window and a relatively coarse spectral resolution of approx. 1.3–1.4 nm (resolving power of 1400–1600). Thompson et al. (2016) demonstrated the ability to resolve and quantify methane ($CH_4$) plumes, posing a similar remote sensing challenge as $CO_2$, using data from the space-borne Hyperion imaging spectrometer, with a spectral and spatial resolution of 10 nm (resolving power around 230) and 30 m, respectively. Observation

of emission plumes, from plume detection to enhancement quantification and flux estimation, using imaging spectroscopy with a single narrow spectral window and a spectral resolution as coarse as 5 to 10 nm (resolving power around 200–500) has further been repeatedly demonstrated using airborne imaging spectroscopy data for both $CO_2$ (Dennison et al., 2013; Thorpe et al., 2017) and $CH_4$ (Thorpe et al., 2014; Thompson et al., 2015; Thorpe et al., 2016, 2017; Jongaramrungruang et al., 2019). For an airborne instrument primarily dedicated to the quantitative imaging of $CH_4$, but also $CO_2$ plumes, Thorpe

et al. (2016) proposed a single spectral window and a spectral resolution of 1.0 nm (resolving power around 2000–2400), again coarse enough to reach a spatial resolution on the order of 10–100 m. The commercial instrument GHGSat-D operated by the Canadian company GHGSat Inc. was launched in 2016 as a demonstrator for a satellite constellation concept targeting the detection of $CH_4$ plumes from individual point sources within selected $\approx 10 \times 10\,\mathrm{km^2}$ target regions at a spectral and spatial resolution of 0.1 nm (resolving power around 16 000) and 50 m, respectively (**?**). Varon et al. (2019) recently showed how

anomalously large $CH_4$ point sources can be discovered with GHGSat-D observations.

Given the results from previous studies and the technology at hand, we are confident that the proposed instrument concept presented here could be realised and that it would be an important complement to the fleet of current and planned space-borne $CO_2$ instruments, allowing for the routine quantitative monitoring of $CO_2$ emissions from large and medium-sized power plants. The proposed instrument concept would also serve as a good complement and companion to CO2M, by targeting also

medium-sized power plants and providing high-resolution images with finer $CO_2$ plume structures. The added value of such an instrument would be of interest, both in terms of advancing science as well as in providing independent emission estimates that could be used to verify reported $CO_2$ emission rates at facility level and inform policy makers on the progress of reducing man-made $CO_2$ emissions. The proposed instrument concept is described in Sect. 2, followed by a description of the instrument noise model in Sect. 3. A global performance assessment addressing instrument noise and the errors introduced by atmospheric

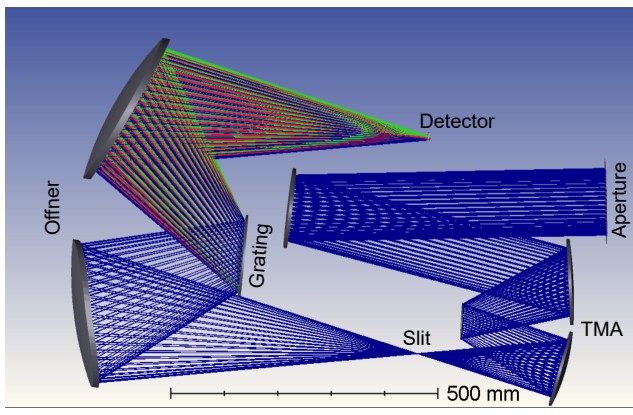

**Figure 2.** Ray-tracing diagram of the preliminary optical design assuming a Three-Mirror-Anastigmat (TMA) telescope combined with an Offner-type spectrometer.

aerosol is presented in Sect. 4. The ability to resolve single $CO_2$ emission plumes at urban scale is further simulated in Sect. 5. A short summary and our concluding remarks are finally presented in Sect. 6.

## 2    Mission and instrument concept

The instrument concept presented in this paper is based on a space-borne push-broom imaging grating spectrometer, measuring spectra of reflected solar radiation in one single SWIR spectral window, from which the column-averaged dry-air mole fraction of $CO_2$ ($XCO_2$) can be retrieved. With an expected instrument mass of approx. 90 kg, it is suitable for the deployment on small satellite buses. Since the proposed instrument is targeting the quantification of localized $CO_2$ emissions from e.g. coal-fired power plants, a high spatial resolution of $50 \times 50 \, \text{m}^2$ is envisaged. The instrument is designed to fly in a sun-synchronous orbit at an altitude of 600 km and a local equatorial crossing time at 13:00. This orbit is chosen in order to have a well developed boundary layer at overpass together with good radiometric performance (high SNR).

The preliminary optical design assumes a circular aperture (15 cm in diameter) and is based on a Three-Mirror-Anastigmat (TMA) telescope, combined with an Offner-type spectrometer, as shown in Fig. 2. The optic system relies on metal-based mirrors and is designed as an athermal configuration for a wide temperature range onboard the satellite. The three mirrors of the TMA are standard aspheres aligned on a single optical axis. The efficiency of the optical bench (throughput), including e.g. transmittance and grating efficiency, is estimated to 0.48 and the f-number ($f_{\text{num}}$), equal to the ratio of focal length to aperture diameter, amounts to 2.4. The dispersed electromagnetic radiation is focused onto a two-dimensional array detector that captures the spatial across-track dimension as well as the spectral dimension of the incoming radiation. A detector with a pixel area of $900 \, \mu\text{m}^2$ and a quantum efficiency of 0.8 is assumed for this study. The quantum efficiency depends on the wavelength, but is for now assumed constant for both spectral windows. These values are in line with typical values for a state-of-the-art detector.

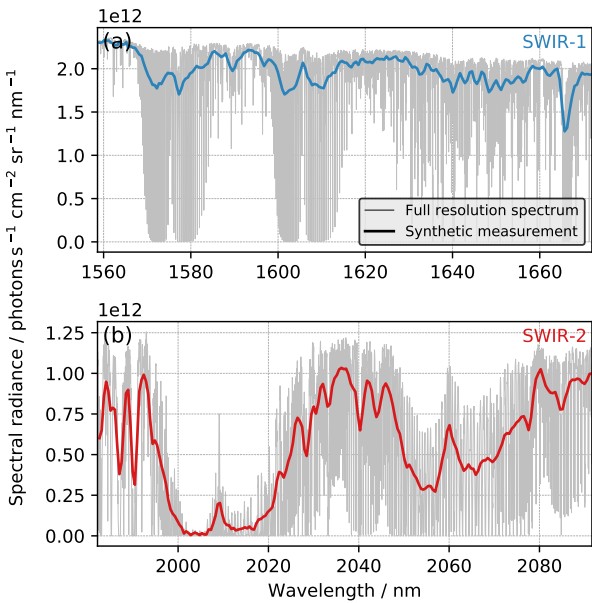

**Figure 3.** Simulated synthetic measurements of spectral radiances for the two spectral set-ups near 1600 nm (SWIR-1) **(a)** and 2000 nm (SWIR-2) **(b)** for our reference scene with surface albedo 0.1 and SZA = 70°. Thin grey lines show corresponding spectral radiances at approx. 0.003 nm spectral resolution.

In order to reach a sufficient SNR, the proposed spatial resolution only allows for a relatively coarse spectral resolution. Wilzewski et al. (2020) used spectrally degraded GOSAT soundings to demonstrate the capability of retrieving $XCO_2$ from a single spectral window at such a coarse spectral resolution using a spectral set-up (in terms of spectral range, resolution and oversampling ratio) compact enough to fit onto 256 detector pixels. They evaluate two alternative spectral set-ups covering 5   the spectral ranges 1559–1672 nm (hereafter also referred to as SWIR-1) and 1982–2092 nm (hereafter also referred to as SWIR-2), each with a spectral resolution (full-width half-maximum (FWHM) of the instrument spectral response function) of 1.37 nm and 1.29 nm, respectively and an oversampling ratio of three. The resolving power of the SWIR-1 and and SWIR-2 set-ups amounts to approx. 1200 and 1600, respectively. For optics design reasons, we use a spectral oversampling ratio of 2.5 in this study, resulting in a spectral sampling distance of approx. 0.55 and 0.52 nm for SWIR-1 and SWIR-2, respectively. 10   Simulated synthetic measurements of spectral radiances for the two prospective spectral set-ups are shown in Fig. 3, assuming a Gaussian instrument response function with FWHM of 1.37 nm and 1.29 nm, respectively, as proposed by Wilzewski et al. (2020). The SWIR-1 window (Fig. 3a) exhibits two weak $CO_2$ absorption bands around 1568–1585 nm and 1598–1615 nm and has the advantage of a stronger top-of-atmosphere (TOA) signal due to higher solar irradiance and surface albedo at these wavelengths. It also allows for the simultaneous retrieval of $CH_4$ using the $CH_4$ absorption band near 1666 nm. The SWIR-2 15   window, on the other hand, exhibits two stronger $CO_2$ absorption bands around 1995–2035 nm and 2045–2080 nm and has higher sensitivity to atmospheric aerosol that can potentially be exploited during the $XCO_2$ retrieval (Wilzewski et al., 2020).

**Table 1.** Mission and instrument design parameters of the proposed space-borne $CO_2$ monitoring instrument concept.

| | |
|---|---|
| Orbit | 600 km, sun-synchronous |
| Mass / kg | 90 |
| Swath / km | 50 |
| Spatial resolution / $m^2$ | $50 \times 50$ |
| Spectral range / nm | 1559–1672 **or** 1982–2092 |
| FWHM (2.5 pix) / nm | 1.37 **or** 1.29 |
| Resolving power / - | 1200 **or** 1600 |
| Aperture diameter / cm | 15.0 |
| f-number ($f_{num}$) / - | 2.4 |
| Optical efficiency ($\eta$) / - | 0.48 |
| Integration time ($t_{int}$) / ms | 70 |
| Detector pixel area ($A_{det}$) / $\mu m^2$ | 900 |
| Quantum efficiency ($Q_e$) / $e^-$ photon$^{-1}$ | 0.8 |
| Dark current ($I_{dc}$) / fA pix$^{-1}$ s$^{-1}$ | 1.6 |
| Readout-noise / $e^-$ | 100 |
| Quantization noise / $e^-$ | 40 |

Wilzewski et al. (2020) showed similar performance for SWIR-1 and SWIR-2, respectively, but suspect SWIR-2 to be the favourable spectral set-up given the stronger $CO_2$ absorption bands, the ability to account for particle scattering and the lower radiance SNR required to reach sufficiently small $XCO_2$ noise errors. In this paper, we further investigate the performance of the two spectral set-ups in order to finally conclude on the more suitable one given the preliminary instrument design and

realistic instrument SNR assumed here.

    The instrument is designed to have a radiance SNR of 100 at the continuum for a reference scene with a Lambertian surface albedo of 0.1 and solar zenith angle (SZA) of 70°. Given the altitude of 600 km and the corresponding orbital velocity of 7562 m s$^{-1}$, the instrument traverses along one 50 m ground pixel in approx. 7.2 ms. The amount of photons collected over the course of 7.2 ms is, however, not enough to reach a SNR of 100. To increase the SNR, we suggest to increase the integration

time to 70 ms. This would normally lead to elongated ground pixels (approx. $50 \times 500$ $m^2$), but by using forward motion compensation (FMC), the instrument can be periodically altered in the along-track direction, such that each ground pixel is sampled for a time period longer than the actual satellite overpass time (see e.g. Sandau, 2010; Abdollahi et al., 2014). FMC has the evident drawback that the coverage along the satellite track is discontinuous, since no data are sampled when the instrument returns to the starting forward position. A second disadvantage is the geometrical distortion of the ground pixels, that increases

with the maximum off-nadir angle. The baseline design assumes 1000 measurements to be made in the along-track dimension

for each FMC repetition, leading to off-nadir angles up to approx. $20°$. Further assuming a 1000 detector pixels in the spatial dimension would consequently result in observed tiles on the order of $50 \times 50\,\mathrm{km}^2$.

Table 1 summarizes the preliminary mission concept and instrument design parameters assumed for this study. It should be clear that this is a preliminary baseline design used to demonstrate the $CO_2$ monitoring abilities and added value of the proposed instrument concept. Alternative instrument designs will be further investigated and the exact instrument design will most likely be subject to change before the instrument would be realized. The continuum SNR for our reference scene should, nevertheless, remain at roughly 100, ensuring a similar performance as presented in this paper.

## 3 Instrument noise model

To assess the performance of the proposed instrument concept w.r.t. retrieving $XCO_2$ and resolving localized $CO_2$ emissions, the expected instrument noise levels that accompany the measurements have to be quantified. To this end a numerical instrument noise model that calculates the instrument's SNR is developed, following a similar approach as e.g. Bovensmann et al. (2010) and Butz et al. (2015). The SNR is given by

$$\mathrm{SNR} = \frac{S}{\sigma_{\mathrm{tot}}}, \tag{1}$$

where $S$ is the signal, i.e. the number of photons emerging from a $50 \times 50\,\mathrm{m}^2$ ground pixel that generate a charge in the detector and $\sigma_{\mathrm{tot}}$ is the corresponding instrument noise. The signal $S$ is calculated as

$$S = L_\lambda \cdot \frac{\pi \cdot A_{\mathrm{det}}}{4 \cdot f_{\mathrm{num}}^2} \cdot \eta \cdot Q_e \cdot \Delta\lambda \cdot t_{\mathrm{int}}, \tag{2}$$

where $L_\lambda$ is the simulated reflected solar spectral radiance at the telescope, $A_{\mathrm{det}}$ the detector pixel area, $f_{\mathrm{num}}$ the instrument's f-number, $\eta$ the efficiency of the optical bench, $Q_e$ the detector's quantum efficiency, $\Delta\lambda$ the wavelength range covered by a single detector pixel and $t_{\mathrm{int}}$ the integration time between the detector pixel read-outs. Following the thin lens equation (for large distances between lens and object) and the magnification formula, the term $\frac{\pi \cdot A_{\mathrm{det}}}{4 \cdot f_{\mathrm{num}}^2}$ can also be expressed as $A_{\mathrm{ap}} \cdot \Omega$, where $A_{\mathrm{ap}}$ is the area of the aperture ($= \pi r^2$ with $r = 7.5\,\mathrm{cm}$) and $\Omega$ the instrument's solid angle i.e. the squared ratio of the ground sampling distance ($50\,\mathrm{m}$) over the orbit altitude ($600\,\mathrm{km}$). Apart from $L_\lambda$ that is calculated for each scene using a forward radiative transfer model, all quantities in Eq. 2 and their corresponding values were introduced in Sect. 2.

The total noise $\sigma_{\mathrm{tot}}$ in Eq. 1 accounts for the noise contribution from five separate instrument noise sources

$$\sigma_{\mathrm{tot}} = \sqrt{\sigma_{\mathrm{ss}}^2 + \sigma_{\mathrm{bg}}^2 + \sigma_{\mathrm{dc}}^2 + \sigma_{\mathrm{ro}}^2 + \sigma_{\mathrm{qz}}^2}, \tag{3}$$

where $\sigma_{\mathrm{ss}} = \sqrt{S}$ is the signal shot noise, $\sigma_{\mathrm{bg}}$ is the noise due to thermal background radiation incident on the detector, $\sigma_{\mathrm{dc}}$ is the noise due to dark current in the detector, $\sigma_{\mathrm{ro}}$ is the noise upon detector read-out and $\sigma_{\mathrm{qz}}$ the quantization noise that arises when the analog signal is digitized. The thermal background signal per detector pixel is approximated as

$$S_{\mathrm{bg}} = A_{\mathrm{det}} \cdot Q_e \cdot t_{\mathrm{int}} \cdot E_{\mathrm{BB}}, \tag{4}$$

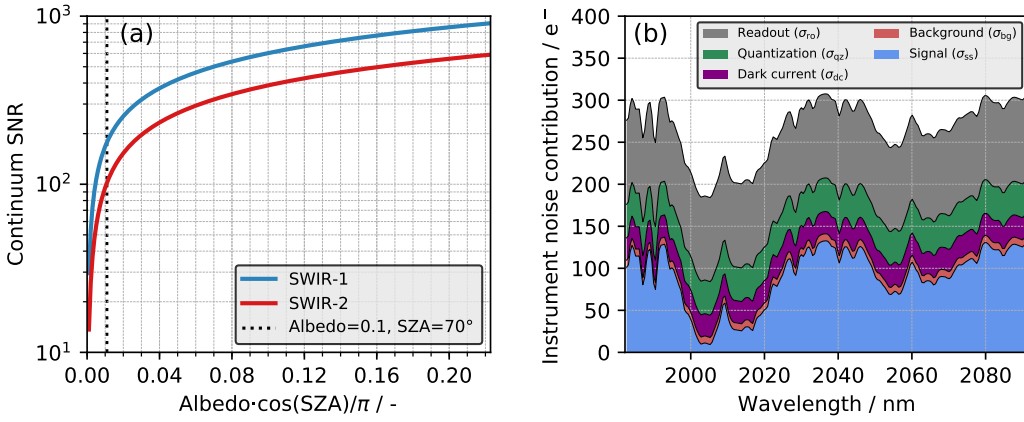

**Figure 4. (a)** Continuum SNR as a function of scene brightness (Albedo · cos(SZA)/$\pi$) for the SWIR-1 and SWIR-2 spectral set-ups, with the dotted line indicating the brightness of our reference scene. The highest scene brightness (approx. 0.22) represents a bright scene with albedo=0.7 and SZA = $0°$ **(b)** Instrument noise contributions for a simulated SWIR-2 spectrum.

where $E_{\mathrm{BB}}$ is the thermal black-body irradiance incident on the detector. $E_{\mathrm{BB}}$ is determined by integrating the black-body spectral radiance $L_{\lambda,\mathrm{BB}}(T_{\mathrm{bg}})$ emitted by the background over the detector's cut-off wavelengths $\lambda_1$ and $\lambda_2$ and hemispheric opening angle

$$E_{\mathrm{BB}} = \pi \int_{\lambda_1}^{\lambda_2} L_{\lambda,\mathrm{BB}}(T_{\mathrm{bg}})d\lambda. \tag{5}$$

For this study, detector cut-off wavelengths of 900 and 2500 nm are assumed and the background temperature $T_{\mathrm{bg}}$ is estimated to 200 K. The thermal background noise is then calculated as $\sigma_{\mathrm{bg}} = \sqrt{S_{\mathrm{bg}}}$. Similarly, the dark current noise is given by $\sigma_{\mathrm{dc}} = \sqrt{S_{\mathrm{dc}}}$, where $S_{\mathrm{dc}} = I_{\mathrm{dc}} \cdot t_{\mathrm{int}} \cdot Q$ is the per-pixel detector signal due to dark current. While $Q = 6.242 \cdot 10^{18}$ electrons Coulomb$^{-1}$ is constant, the dark current $I_{\mathrm{dc}}$ strongly depends on the detector's operating temperature and is estimated to $1.6\,\mathrm{fA\,pix}^{-1}\,\mathrm{s}^{-1}$ (assuming 150 K detector temperature), yielding a dark current signal of approx. 10 000 electrons (e$^-$) per detector pixel and

second. Finally, the read-out noise ($\sigma_{\mathrm{ro}}$) and quantization noise ($\sigma_{\mathrm{qz}}$) are estimated to 100 and 40 e$^-$, respectively. These noise levels are preliminary estimates used to test and evaluate the instrument concept, but are comparable to those of state-of-the-art detectors for space applications.

     Figure 4a shows the continuum SNR (calculated with Eqs. (1)–(5)) as a function of the scene brightness for the two prospective spectral set-ups SWIR-1 and SWIR-2. The scene brightness describes the conversion from incident solar irradiance to

reflected solar radiation and is calculated as the product of the surface albedo and the cosine of the SZA, divided by $\pi$, hence assuming a Lambertian surface. For the reference scene (albedo = 0.1, SZA = $70°$), the continuum SNR is approx. 180 and 100 for SWIR-1 and SWIR-2, respectively. The consistently higher SNR for SWIR-1, compared to SWIR-2, is mainly the result of higher solar radiance (see Fig. 3) as well as generally higher surface albedo (see e.g. Fig. 7 in Butz et al. (2009)) in SWIR-1. Looking at the individual contributions from the different instrument noise sources in Fig. 4b, it is clear that the

readout noise and signal shot noise are the major contributors, whereas the noise arising from quantization errors, dark current and thermal background radiation has a small or even negligible contribution in comparison. The signal shot noise is, however, smaller than the dark current, read-out noise and quantization noise inside the $CO_2$ absorption bands, where the signal, and hence the signal shot noise, decreases. Note that all noise terms, except for the signal shot noise $\sigma_{ss}$, are constant.

## 4 Generic performance evaluation

In this section we conduct a first performance evaluation of the proposed instrument concept by assessing the $XCO_2$ retrieval errors expected on a global scale. Such errors arise due to instrument noise and because of inadequate knowledge about the light path through the atmosphere due to scattering aerosol and cirrus particles. For this purpose we use a global trial ensemble with a large collection of geophysical scenarios with varying atmospheric gas concentrations, meteorological conditions, surface albedo, SZA as well as aerosol and cirrus compositions, that can be expected to be observed by a polar orbiting instrument. The same methodology and dataset have been used in several previous studies to assess the greenhouse gas retrieval performance of different satellite instruments (Butz et al., 2009, 2010, 2012, 2015).

The global trial ensemble contains geophysical data representative for the months of January, April, July and October. Atmospheric gas concentrations stem from the CarbonTracker model ($CO_2$ for the year 2010, Peters et al., 2007), the Tracer Model 4 ($CH_4$ for the year 2006, Meirink et al., 2006) and the ECHAM5-HAM model ($H_2O$, Stier et al., 2005). Surface albedo data, representative for the SWIR-1 and SWIR-2 windows, respectively, stem from the MODIS (Moderate Resolution Imaging Spectroradiometer) MCD43A4 product (Schaaf et al., 2002). Aerosol optical properties are calculated (assuming Mie scattering) for an aerosol size distribution, superimposed from seven log-normal size distributions and five chemical types at 19 vertical layers, as provided by the ECHAM5-HAM model (Stier et al., 2005). Cirrus optical properties are calculated for randomly oriented hexagonal columns and plates following the ray tracing model of Hess and Wiegner (1994) and Hess et al. (1998). In total the global trial ensemble consists of approx. 10 000 scenes with $XCO_2$ ranging from 340 to 400 ppm with an average of 382 ppm, albedo ranging from 0 to 0.7 with an average of 0.13 (SWIR-2 window), aerosol optical thickness (AOT) ranging from 0 to 1.0 with an average of 0.05 (SWIR-2 window) and cirrus optical thickness (COT) ranging from 0 to 0.8 with an average of 0.13 (SWIR-2 window). Thus, the global trial ensemble contains challenging scenes with scattering loads that would be filtered out by current satellite retrievals, such as those applied to OCO-2 and GOSAT data which typically screen scenes with scattering optical thickness greater than 0.3 (at the $O_2$A band around 760 nm). All data in the global trial ensemble are re-gridded to a spatial resolution of approx. $2.8° \times 2.8°$. This is, of course, much coarser than the envisaged $50 \times 50\,m^2$, but for investigating the propagation of instrument noise into the target quantity $XCO_2$ on a global scale, this dataset serves its purpose. See previous studies (e.g. Butz et al., 2009, 2010) for further details on the content of the global trial ensemble.

The geophysical data for each scene are fed to the radiative transfer software RemoTeC (Butz et al., 2011; Schepers et al., 2014) in order to simulate corresponding synthetic measurements. The measurement noise is calculated by propagating the instrument's SNR (Sect. 3) into a statistical error estimate according to the rules of Gaussian error propagation (Rodgers,

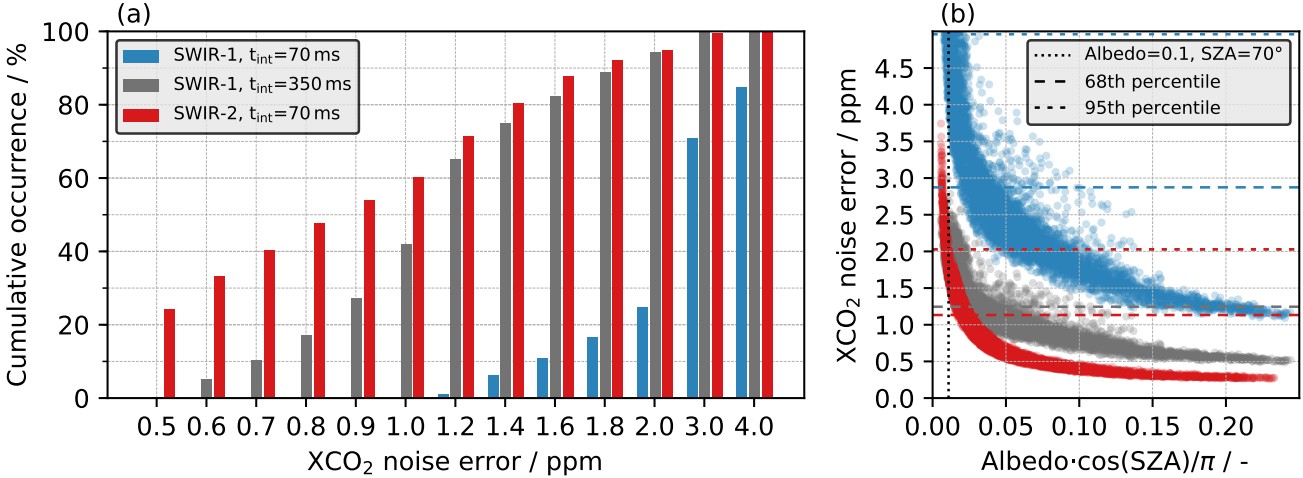

**Figure 5. (a)** Cumulative distribution of the estimated $XCO_2$ noise errors arising from instrument noise for all scenes in the global ensemble. **(b)** $XCO_2$ noise errors as a function of surface brightness, with the dotted line indicating our reference scene and the dashed lines indicating the 68th and 95th percentiles of the $XCO_2$ noise errors for the different spectral set-ups. Note that the red and grey lines for the 95th percentile overlap at 2.03 ppm. Marker colors in **(b)** correspond to those in **(a)**. Both panels show the results for the SWIR-1 (blue) and SWIR-2 (red) set-ups as well as for an alternative SWIR-1 (grey) set-up for comparison (see text for details).

2000). Simulations are conducted globally for the 16th day of each of the four months January, April, July and October, hence covering SZA conditions ranging from 0 to 86 degrees.

By retrieving $XCO_2$ from the simulated synthetic spectra, the range of $XCO_2$ retrieval errors that can be expected with the proposed instrument concept can be estimated, as can the ability to account for atmospheric aerosol. The RemoTeC retrieval

algorithm (e.g. Butz et al., 2011) is based on a Philipps–Tikhonov regularization scheme (Phillips, 1962; Tikhonov, 1963) that uses the first-order difference operator as a side-constraint to retrieve the $CO_2$ partial column profiles, from which $XCO_2$ can be determined. Additional retrieval parameters are the total column concentrations of $H_2O$ and $CH_4$ (only for SWIR-1), surface albedo (as second-order polynomial), spectral shift, solar shift and, possibly, information on scattering aerosol. Here we assume knowledge about the airmass (needed to calculate $XCO_2$), in reality meteorological and topography data would be

required to estimate the airmass.

### 4.1 Instrument noise induced $XCO_2$ errors

In a first step, we assess $XCO_2$ retrieval errors that are induced by instrument noise. To this end, for now, we neglect scattering by aerosol and cirrus. These so-called non-scattering simulations assume no scattering particles to be present in the atmosphere and simply compute the transmittance along the geometric light path (Rayleigh scattering is included).

Figure 5a shows the cumulative distribution of the random $XCO_2$ noise error, i.e. the instrument noise propagated into $XCO_2$ uncertainties via Gaussian error propagation. Furthermore, Fig. 5b shows the $XCO_2$ noise error for each simulated

scene as a function of the corresponding scene brightness. The noise errors are significantly smaller for the SWIR-2 set-up (red) when using the proposed integration time $t_{int}$ of 70 ms. The red dashed lines in Fig. 5b show that on average 68 % and 95 % ($1\sigma$ and $2\sigma$ respectively) of the retrievals have noise errors of less than approx. 1.1 and 2.0 ppm, respectively. For the SWIR-1 set-up (blue), the corresponding numbers are 2.9 and 5.0 ppm. For the SWIR-2 set-up, only retrievals over scenes that are darker than our reference scene (albedo = 0.1, SZA = 70°) are expected to have instrument noise induced errors larger than approx. 2 ppm. For comparison, and as a reference, we also investigate how much the integration time has to be increased for the SWIR-1 set-up, in order to reach a SNR sufficient to yield $XCO_2$ noise errors comparable to those obtained with the SWIR-2 set-up. We find that with the preliminary instrument design assumed here, the integration time has to be increased to at least 350 ms (i.e. by a factor five) for SWIR-1 (grey) in order to reach a similar performance.

Despite the advantage of being able to retrieve $XCH_4$ alongside $XCO_2$ using the SWIR-1 set-up, the much longer integration time required to reach sufficiently low $CO_2$ noise errors is not feasible for the purpose of the proposed instrument concept. Hence, we conclude that the SWIR-2 set-up is superior for the passive satellite based $CO_2$ monitoring instrument proposed in this paper. Consequently, the remainder of this paper is limited to the SWIR-2 set-up, covering the spectral range 1982–2092 nm with a spectral resolution (FWHM) of 1.29 nm, resolving power around 1600 and a spectral sampling distance of 0.52 nm.

## 4.2 Aerosol induced $XCO_2$ errors

Atmospheric aerosol and cirrus particles modify the light path of the reflected solar radiation to a certain degree, depending on the particle abundance, optical properties, height and surface albedo. Consequently, this can cause large errors in the retrieved $XCO_2$ if the effect of $CO_2$ absorption and particle scattering on the measured reflected solar radiation cannot be adequately separated during the retrieval process. In this section the ability to account for atmospheric aerosol and cirrus during the retrieval is investigated by including scattering by atmospheric particles in the simulation of the synthetic measurements as well as in the corresponding $XCO_2$ retrievals. This is done by using a more complex forward model and representation of the aerosol and cirrus particles when simulating the spectra, and a comparatively simple representation and forward model for the corresponding retrievals. More precisely, the full physical representation of vertical profiles of hexagonal cirrus particles and spherical aerosol particles of the five chemical types characterized by the seven log-normal size distributions with known micro-physical properties for each aerosol and cirrus particle type is used when simulating the synthetic measurement for each scene in the global trial ensemble. On the contrary, only three aerosol parameters are fitted during the corresponding retrieval: the total column number density, the size parameter of a single mode power-law size distribution and the center height of a Gaussian height distribution. Such differences in the aerosol/cirrus representation lead to forward model errors that, alongside the instrument noise induced errors, propagate into the retrieved quantity $XCO_2$. Previous studies have shown that this approach gives a good approximation of how well a satellite sensor can account for scattering by atmospheric aerosol while retrieving target gas concentrations (e.g. Butz et al., 2009, 2010).

Figure 6a shows the difference between the $XCO_2$ retrieved ("retr") from the synthetic measurements and the corresponding "true" $XCO_2$ used as input to simulate these synthetic measurements. This deviation from the truth, contains information on

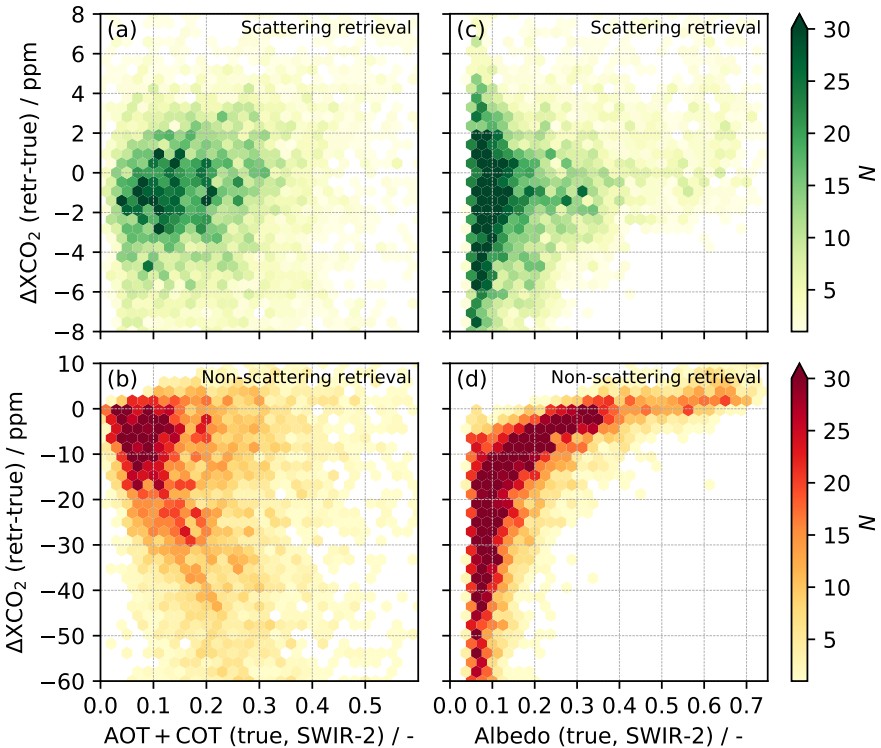

**Figure 6. Left panels** $XCO_2$ retrieval errors as a function of the total particulate optical thickness $AOT + COT$ for scattering **(a)** and non-scattering **(b)** RemoTeC retrievals. **Right panels** $XCO_2$ retrieval errors as a function of SWIR-2 surface albedo for scattering **(c)** and non-scattering **(d)** RemoTeC retrievals. $N$ denotes the number of retrievals. Note the different ranges of the y-axes in the upper and lower panels.

both random instrument noise error (Sect. 4.1) and systematic errors arising from insufficient modelling of the aerosol and cirrus properties. For comparison, Fig. 6b shows the corresponding results achieved when using a non-scattering retrieval, i.e. where the scattering by atmospheric aerosol and cirrus, now present in the atmosphere and the simulated synthetic spectra, is neglected (similar to Sect. 4.1). The retrieval errors are strongly reduced when the RemoTeC retrieval algorithm accounts
5 for the scattering by atmospheric aerosol. When scattering is considered, half of the $XCO_2$ retrievals deviate from the true abundance by less than 2.5 ppm while two thirds of the retrievals deviate by less than 4 ppm (approx. 1 %), with no clear error-correlation with the optical thickness of the scattering particles. For the non-scattering retrieval, the corresponding numbers are 16 and 28 ppm, with a mean bias of -25 ppm that increases with optical thickness, exposing the necessity of accounting for atmospheric aerosol and cirrus when retrieving the $XCO_2$.
10 Scattering particles can modify the light path and hence the $XCO_2$ retrieval in primarily two ways. Firstly, an elevated layer of aerosol or cirrus will scatter parts of the incoming solar radiation towards the observing sensor at a higher altitude compared to the Earth's surface, leading to a reduced light path. Secondly, aerosol and cirrus will extend the light path to some degree as a

result of multiple scattering between scattering particles and the surface. Such modifications of the light path will be understood as either too low (overall reduced light path) or too high (overall extended light path) $CO_2$ concentrations in the atmosphere if scattering cannot be accounted for in the retrieval. Which effect is dominating, is primarily driven by the surface albedo. This is visualized in Fig. 6d that shows the difference between retrieved and true $XCO_2$ as a function of the surface albedo when scattering by aerosol and cirrus is neglected in the retrieval. Over darker surfaces, where the effect of multiple-scattering between aerosol and surface is limited, aerosol and cirrus particles scattering the incoming solar radiation towards the sensor higher up in the atmosphere becomes the dominating effect, leading to a reduced light path and underestimation of the $XCO_2$. Over brighter surfaces, where the effect of multiple scattering becomes dominant, the non-scattering retrieval is more likely to overestimate the $CO_2$ abundance, because the loss of radiation due to an extended light path, resulting from the multiple scattering, is assumed to be caused by more absorbing $CO_2$ molecules in the atmosphere. Fig. 6c shows the difference between retrieved and true $XCO_2$ as a function of the surface albedo when scattering by aerosol and cirrus is accounted for when retrieving $XCO_2$ from the synthetic measurements of the proposed satellite concept. It is clear that when aerosol properties are retrieved alongside the $CO_2$ abundance, the curve-shaped relationship between the $XCO_2$ error and surface albedo vanishes with no clear error-correlation other than that $XCO_2$ errors increase with decreasing albedo (and thus SNR). Note that errors arising from the Lambertian albedo assumption (BRDF (Bidirectional Reflectance Distribution Function) effects) are neglected in the scattering simulations.

Although layers of aerosol and cirrus can be partly accounted for in the retrieval, scenes with thicker clouds and aerosol layers will have to be identified and filtered out in the data processing chain. Such a cloud filter could exploit the different optical depths of the two $CO_2$ bands in the SWIR-2 window by retrieving $XCO_2$ from the two $CO_2$ bands independently (assuming a non-scattering atmosphere) and filter for discrepancies.

## 5 Performance evaluation for an urban case study

While the previous section assessed $XCO_2$ errors for the range of geophysical conditions expected to be encountered on a global scale, this section evaluates the $CO_2$ monitoring capabilities at urban scale using high-resolution $CO_2$ concentration and surface albedo data. Similar to Sect. 4, the high-resolution data are used to simulate synthetic measurements, from which synthetic $XCO_2$ abundances can be retrieved in order to make a first assessment of the $CO_2$ monitoring ability of the proposed instrument concept in terms of resolving $CO_2$ emission plumes.

### 5.1 Datasets

#### 5.1.1 $CO_2$ concentration field from the Hestia dataset

To compute a high-resolution three-dimensional field of $CO_2$ concentrations to be used as input for the radiative transfer simulations, annual estimates of fossil fuel $CO_2$ emissions for the city of Indianapolis in the year 2015 are used. These data are generated by the Hestia Project (Gurney et al., 2012, 2019) where the fossil fuel $CO_2$ emissions are quantified in urban areas

down to the scale of individual buildings and streets using a bottom-up approach. The results for the city of Indianapolis are gridded and archived at a spatial resolution of $200 \times 200\,\text{m}^2$. For this study, however, the Hestia Project dataset was gridded to $50 \times 50\,\text{m}^2$ via request to the Hestia research team in order to match the envisaged spatial resolution of the proposed instrument concept. The fossil fuel $CO_2$ emission rates for Indianapolis at $50 \times 50\,\text{m}^2$ resolution can be seen in Fig. 7a. $CO_2$ emissions

from different sources and sectors like e.g. road traffic and point sources (single yellow pixels) can be seen. There is also an apparent emission gradient with stronger emissions in the city center and weaker emissions towards the suburbs. Hence, the Hestia $CO_2$ emission data for Indianapolis provide a realistic emission scenario for evaluating the $CO_2$ monitoring capabilities of the proposed instrument concept. Moreover, the area of the Hestia domain (approx. $34 \times 33\,\text{km}^2$) is comparable to what the prospective tile size of each observation target area could be.

The Hestia $CO_2$ emission data are used as input to a Gaussian dispersion model in order to compute a three-dimensional $CO_2$ concentration field. For a given $CO_2$ emission rate $Q$ (in $\text{g s}^{-1}$), the $CO_2$ concentration $C$ (in $\text{g m}^{-3}$) at a given position $(x, y, z)$ downwind of the emitter is calculated as

$$
\begin{aligned}
C(x,y,z) = &\frac{Q}{2\pi u \sigma_y \sigma_z} \exp\left(\frac{-y^2}{2\sigma_y^2}\right) \\
&\left[\exp\left(\frac{-(z-h)^2}{2\sigma_z^2}\right) + \exp\left(\frac{-(z+h)^2}{2\sigma_z^2}\right)\right]
\end{aligned}
\tag{6}
$$

where $u$ is the horizontal wind speed in the $x$-direction (along-wind), $h$ is the height of the emitting source (in m above ground

level) and $\sigma_y$ and $\sigma_z$ are the standard deviations of the concentration distribution (in m) in the horizontal across-wind and vertical dimension, respectively. $\sigma_y$ and $\sigma_z$, and hence the spread of the emission plume, depend on the atmospheric instability i.e. the degree of atmospheric turbulence as well as the downwind distance $x$ from the emitting source. Here, we calculate $\sigma_y$ and $\sigma_z$ assuming the Pasquill-Gifford stability class C (slightly unstable atmosphere). Furthermore, a constant wind speed $u = 3\,\text{m s}^{-2}$ and an emitting source height $h = 75\,\text{m}$ (for all sources) are assumed. This model set-up is comparable to similar

studies (e.g. Bovensmann et al., 2010; Dennison et al., 2013).

Downwind $CO_2$ concentrations from each emitting source (pixel) in the Hestia dataset are calculated across an equidistant grid at $50\,\text{m}$ resolution in all dimensions and the contributions from all individual emitting sources (pixels) are subsequently combined to form a three-dimensional $CO_2$ concentration field over Indianapolis. Figure 7b shows the resulting (vertically integrated) two-dimensional field of (noise-less) $XCO_2$ enhancements at $50 \times 50\,\text{m}^2$ spatial resolution over a constant back-

ground with a surface pressure of $1013\,\text{hPa}$. While weaker diffuse sources like streets cannot be identified, the plumes from stronger point sources are clearly pronounced given the high spatial resolution that allows for a detailed mapping of the plumes. For comparison, Fig. 7c shows the corresponding $XCO_2$ enhancements assuming a coarser spatial resolution of $2 \times 2\,\text{km}^2$. Although the stronger plumes can still be identified at the coarser resolution, the $XCO_2$ enhancements are significantly lower and each plume is only sampled by a few pixels. Figure 7d further shows these $XCO_2$ enhancements in more detail for three along-

track excerpts centred at 400, 1500 and 4000 m downwind of the strongest emitter in Indianapolis, with an annual emission rate of $3.24\,\text{MtCO}_2\,\text{yr}^{-1}$ in 2015. The position of the three along-track excerpts are indicated with grey lines in Figs. 7b and 7c. With a spatial resolution of $2 \times 2\,\text{km}^2$, the along-track plume excerpts are only sampled by one pixel each, with a maximum

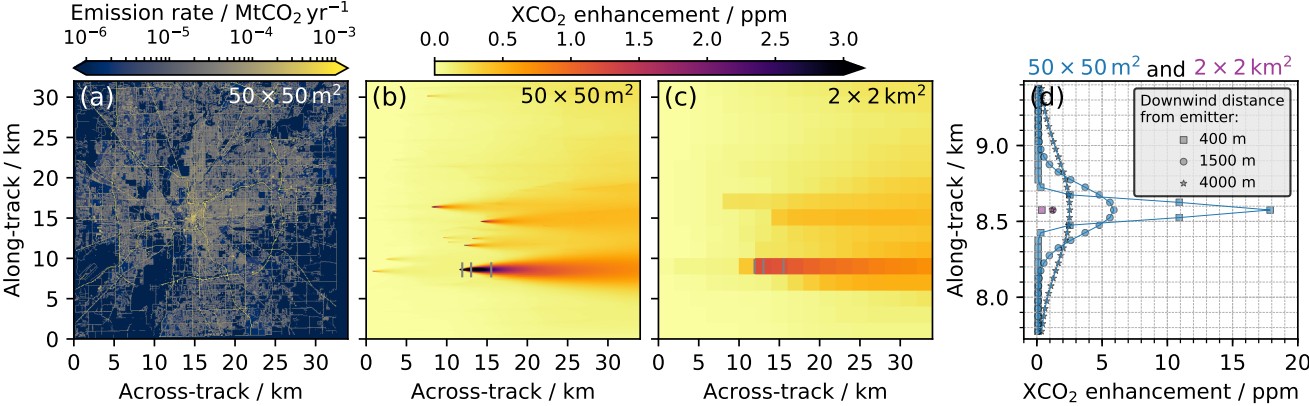

**Figure 7. (a)** Hestia fossil fuel $CO_2$ emission data for Indianapolis in 2015 at $50 \times 50\,\text{m}^2$ spatial resolution. **(b)** Corresponding field of vertically integrated $XCO_2$ enhancements at $50 \times 50\,\text{m}^2$ spatial resolution w.r.t. a constant background, computed using the Hestia $CO_2$ emission data and a Gaussian dispersion model. **(c)** same as **(b)**, but at $2 \times 2\,\text{km}^2$ spatial resolution. **(d)** Per-pixel $XCO_2$ enhancements for three along-track excerpts centred at 400, 1500 and 4000 m downwind of the emitter at $50 \times 50\,\text{m}^2$ and $2 \times 2\,\text{km}^2$ spatial resolution. The respective position of the along-track excerpts are indicated by the small grey lines in **(b)** and **(c)**. The x- and y-dimensions of the Hestia Indianapolis domain are illustrated as hypothetical satellite across-track and along-track dimensions, respectively.

$XCO_2$ enhancement of 1.2 ppm. With the envisaged $50 \times 50\,\text{m}^2$ spatial resolution, however, the plume is sampled by 7, 15 and 29 pixels in the along-track dimension 400, 1500 and 4000 m downwind of the emitter, respectively, with maximum $XCO_2$ enhancements reaching approx. 18, 6 and 3 ppm, respectively. This clearly demonstrates the benefit of an instrument with a high spatial resolution when resolving $CO_2$ emission plumes from space.

### 5.1.2 Surface albedo data from Sentinel-2

To accurately simulate the instrument SNR and hence the measurement noise, it is important to know how large a fraction of the solar radiation incident on the Earth's surface is reflected back towards space. To get realistic estimates of the surface albedo within the Hestia Indianapolis domain, data from the European Sentinel-2 satellites are used. The multi-spectral instrument aboard Sentinel-2 measures the TOA radiance in 13 spectral bands with a spatial resolution ranging from $10 \times 10\,\text{m}^2$ to $60 \times 60\,\text{m}^2$. For this study, we use the Sentinel-2 L1C radiances measured in the spectral band 12 (centred at approx. 2200 nm) at a spatial resolution of $20 \times 20\,\text{m}^2$. The software *Sen2Cor* (ESA, 2018) is employed to compute corresponding L2 surface reflectances from the L1C TOA radiances, through a so-called atmospheric correction.

Surface reflectance data for the month of July 2018 are computed and re-gridded (using nearest neighbour) to the envisaged spatial resolution of $50 \times 50\,\text{m}^2$. The surface reflectance for Sentinel-2 pixels classified as vegetation are scaled by a factor 0.82 in order to account for the generally lower reflectance by vegetation in the SWIR-2 window compared to Sentinel-2's band 12. The scaling factor has been derived using spectral reflectance data from the ECOSTRESS spectral library (Baldridge et al.,

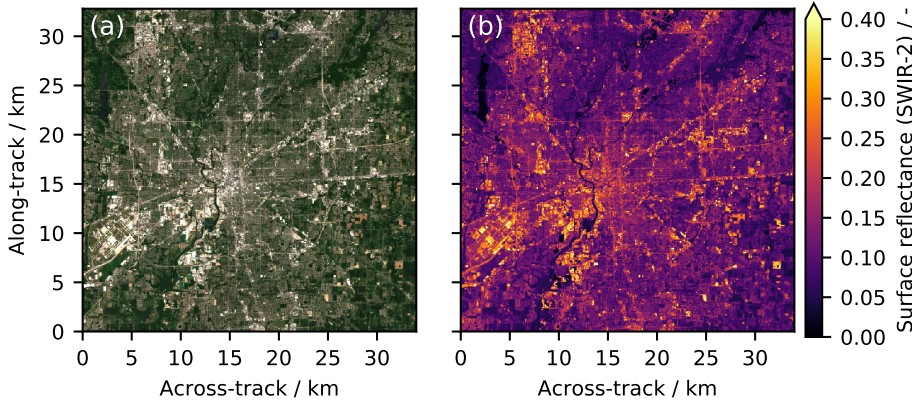

**Figure 8. (a)** Sentinel-2 true color RGB image of Indianapolis (Hestia domain) at $50 \times 50\,\text{m}^2$ spatial resolution derived from Sentinel-2 measurements in July 2018. **(b)** Corresponding surface reflectance data using data from Sentinel-2's spectral band 12 centred at approx. 2200 nm, scaled to the SWIR-2 spectral window. Again, the x- and y-dimensions of the Hestia Indianapolis domain are illustrated as hypothetical satellite across-track and along-track dimensions, respectively.

2009; Meerdink et al.). Figure 8b shows the gridded surface reflectance data for Indianapolis together with a corresponding RGB composite (Fig. 8a), using the Sentinel-2 data from the bands centred at red, green and blue wavelengths, as reference. The scaled and gridded Sentinel-2 surface reflectance data are taken as representative for a constant Lambertian surface albedo within the entire SWIR-2 window. Figure S1 in the supplement shows spectral reflectances in the SWIR spectral region for

various urban materials using data from the Spectral Library of impervious Urban Materials (SLUM; Kotthaus et al., 2014). The small spectral variations within the SWIR-2 spectral window used in this study indicate that the true spectral signatures of the surface albedo could be fitted with sufficient precision using the second order polynomial during the retrieval, and hence the assumption of a constant albedo is reasonable for this study.

The average surface reflectance within the Hestia domain is 0.13. Despite annual variability in surface reflectance, mainly

due to changes in vegetation/crops, this is a value representative throughout most of the year. For comparison, average surface reflectances from the same source for January (snow-free days), April and October 2018, amount to 0.11, 0.17 and 0.11, respectively.

### 5.1.3   Background data from CarbonTracker

Background data, including vertical profiles of $CO_2$, $H_2O$, temperature and pressure, are taken for the 15th of July 2016

from the CarbonTracker CT2017 dataset (Peters et al., 2007, with updates documented at http://carbontracker.noaa.gov). The CarbonTracker CT2017 data over Indianapolis are provided at a spatial resolution of $1° \times 1°$, meaning that the entire Hestia Indianapolis domain is covered by one single CarbonTracker pixel leading to a constant background data field.

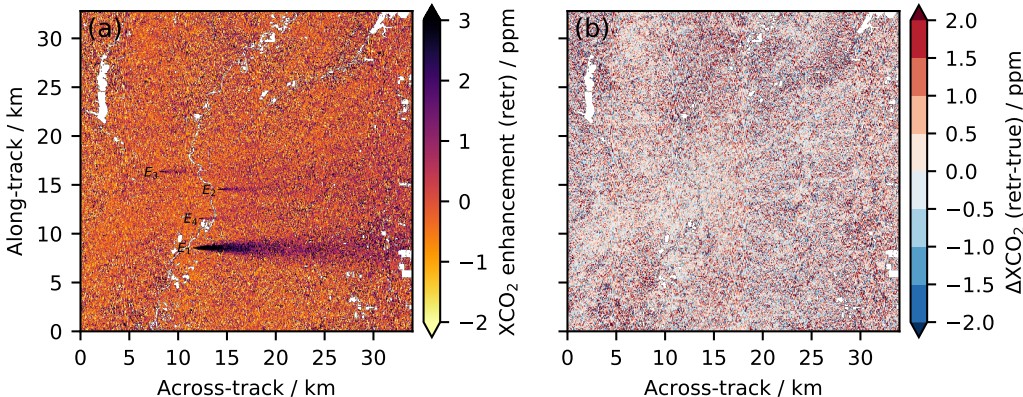

**Figure 9. (a)** $XCO_2$ enhancements w.r.t. the background retrieved from the simulated synthetic measurements over the Hestia Indianapolis domain under non-scattering conditions. Locations of the four strongest point sources are labelled with $E_{1-4}$. **(b)** Corresponding deviations between retrieved ("retr") and true $XCO_2$. Dark scenes with albedo < 0.05 have been filtered out due to unreliable $XCO_2$ retrievals.

## 5.2 Simulated CO₂ plume observations

As in Sect. 4, the above sets of input data are used to simulate synthetic measurements (spectral radiances) and corresponding instrument noise of the proposed instrument concept using the forward model and the instrument noise model (Sect. 3). The SZA is calculated for the given coordinates in the Hestia domain assuming the sun-synchronous orbit described in Sect. 2 and an observation date of July 15, 2018, which translate to a SZA of about 18°. Corresponding $XCO_2$ abundances are then retrieved from the simulated spectral radiances, such that the ability to observe the $CO_2$ emission plumes from the Hestia Indianapolis data can be evaluated. In this first assessment we focus solely on the instrument performance in terms of its $CO_2$ plume quantification capabilities and hence we perform the high-resolution simulations with the expected instrument noise induced errors only, i.e. by assuming a non-scattering atmosphere.

Figure 9a shows the retrieved field of $XCO_2$ enhancements w.r.t. the retrieved background $XCO_2$ over the Hestia domain. The $CO_2$ plume from the strongest point source, $E_1$, with an annual $CO_2$ emission rate of $Q_1 = 3.24 \, \mathrm{MtCO_2 \, yr^{-1}}$, is clearly resolved with local $XCO_2$ enhancements well above 100 ppm close to the emitting source. Although they emit considerably less $CO_2$, the plumes from the second and third strongest point sources, $E_2$ and $E_3$, with annual $CO_2$ emission rates of $Q_2 = 0.55 \, \mathrm{MtCO_2 \, yr^{-1}}$ and $Q_3 = 0.48 \, \mathrm{MtCO_2 \, yr^{-1}}$, respectively, can be clearly separated from the background as well. The plume from the fourth strongest point source, $E_4$, with an annual $CO_2$ emission rate of $Q_4 = 0.32 \, \mathrm{MtCO_2 \, yr^{-1}}$ can also be observed, but is partly obscured by filtered out dark surface areas, where retrieval errors are too high. Plumes from weaker point sources ($\lesssim 0.1 \, \mathrm{MtCO_2 \, yr^{-1}}$) and other sources like e.g. streets and highways cannot be identified given the spatial resolution and instrument noise errors of the proposed instrument.

One concern with high-resolution $CO_2$ remote sensing is the impact of the albedo heterogeneity at urban scale at such a high spatial resolution. For the non-scattering scenario simulated here, the albedo fitted by the retrieval algorithm matches

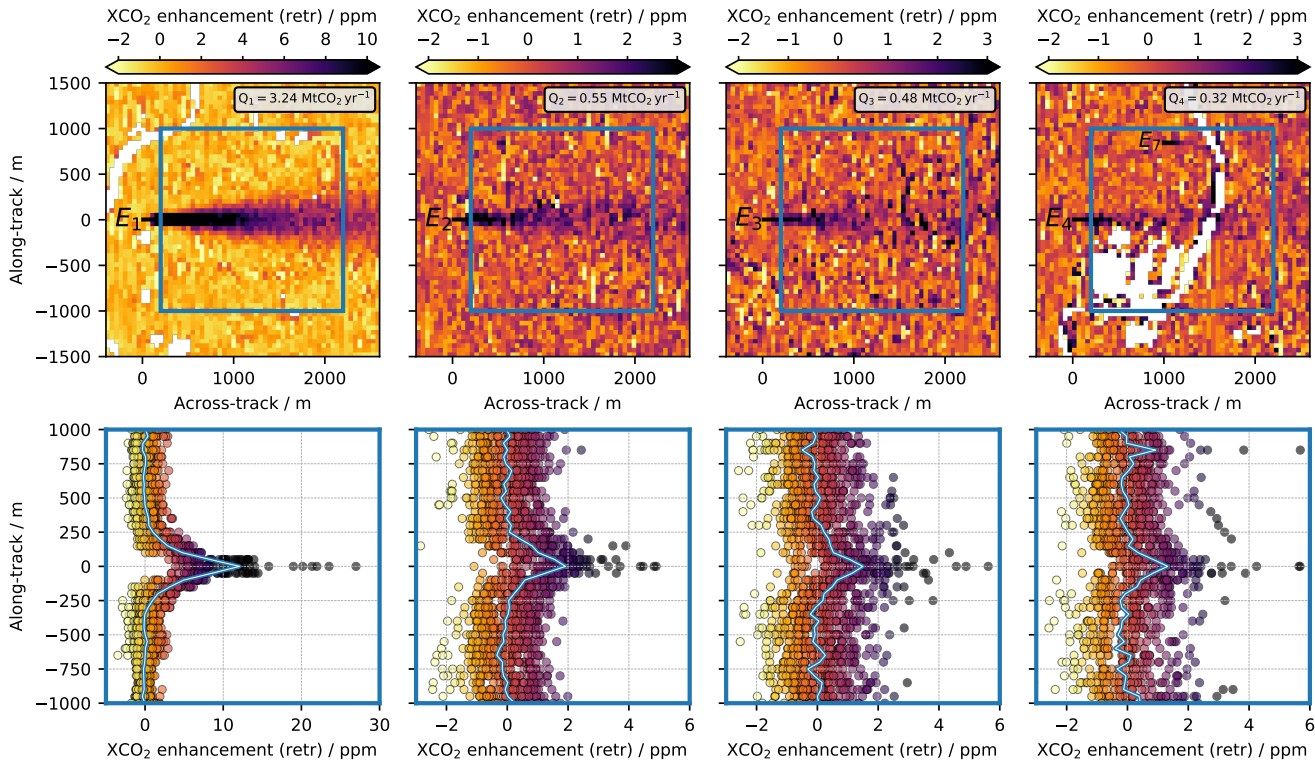

**Figure 10. Upper panels** Retrieved two-dimensional fields of $XCO_2$ enhancements in the vicinity of the four strongest $CO_2$ emitters $E_1$, $E_2$, $E_3$ and $E_4$ within the Hestia Indianapolis dataset. **Lower panels** Corresponding per-pixel (circles) and average (solid lines) along-track $XCO_2$ enhancements within the area 200 to 2200 m downwind and -1000 to 1000 across-wind of the respective emitters. The blue rectangles in the upper panels show the areas from which the corresponding per-pixel and average along-track $XCO_2$ enhancements, depicted in the respective lower panels, are extracted and calculated. The color of the circles follow the color bars in the respective upper panels.

the reference input albedo with an average (absolute) deviation of 0.14 %, and there is consequently no spatial variability in the accuracy of the albedo retrieval that in turn affect the $XCO_2$ retrieval accuracy. There is, however, the evident effect that a higher albedo leads to a higher SNR and hence a generally lower noise error. This is evident from Fig. 9b showing the difference between the retrieved and true $XCO_2$, thus illustrating an instantaneous noise error field that would be expected for a single satellite overpass. Generally, the deviations from the true $XCO_2$ are smaller over areas of brighter surfaces like concrete, whereas the deviations are larger over dark surfaces like forests (see also Fig. 8). The effect of albedo heterogeneity in combination with scattering particles is not addressed in this paper and will have to be analysed in future studies.

Across the entire Hestia domain (but excluding dark scenes with albedo < 0.05) 68 % and 95 % of the $XCO_2$ retrievals deviate from the true $XCO_2$ by less than 1.1 and 2.3 ppm, respectively. This is comparable to the noise error obtained for the global trial ensemble in Sect. 4.1 (1.1 and 2.0 ppm, respectively).

Figure 10 shows close-ups of the simulated $XCO_2$ enhancement field (upper panels) in the vicinity of the four strongest emitters in the Hestia Indianapolis dataset ($E_1 - E_4$ in Fig. 9a), along with the corresponding per-pixel and average along-track $XCO_2$ enhancements (lower panels) for the range 200–2200 m downwind of the respective emitting sources. Enhancements from the 200 m closest to each emitting source are excluded as those scenes could likely be obscured by condensate in a real situation.

The plume of the strongest emitter $E_1$ in Indianapolis with an annual emission rate of $Q_1 = 3.24\,\mathrm{MtCO_2}\,\mathrm{yr}^{-1}$ (Fig. 10, left panels) is clearly resolved. Within the area 200-2200 m downwind of the emitting source (blue square), maximum enhancements exceed 25 ppm, and in total, approx. 200 (60) pixels have enhancements above 4 (8) ppm, representing enhancements of approx. 1 (2) % w.r.t. the background. The average along-track $XCO_2$ enhancement 200-2200 m downwind of the emitting source (blue/white line) reaches 12 ppm. The plumes from the second and third strongest emitters $E_2$ and $E_3$, approx. six times weaker than $E_1$, with annual emission rates of $Q_2 = 0.55$ and $Q_3 = 0.48\,\mathrm{MtCO_2}\,\mathrm{yr}^{-1}$, respectively (Fig. 10, center panels) have considerably lower $XCO_2$ enhancements, but can nevertheless be clearly separated from the background with distinct increments in both per-pixel and average $XCO_2$ enhancements within the area 200–2200 m downwind of the emitters (blue squares). While the background fields varies from approx. -1 to 1 ppm due to instrument noise, the per-pixel plume enhancements vary from approx. 0.5 to 3 ppm, with single enhancements exceeding 4 ppm close to the emitting source. The average along-track $XCO_2$ enhancements 200-2200 m downwind of the emitting sources (blue/white lines) reach 1.9 and 1.5 ppm for $E_2$ and $E_3$, respectively. Despite being partly obscured by filtered out dark surfaces (water), also the plume from the fourth strongest emitter $E_4$, with an annual emission rate of $Q_4 = 0.32\,\mathrm{MtCO_2}\,\mathrm{yr}^{-1}$ (Fig. 10, right panels) can be separated from the background, both when looking at the two-dimensional field and the per-pixel enhancements within the area 200-2200 m downwind of the emitter. With maximum average $XCO_2$ enhancements of at most approx. 1.3 ppm, the proposed instrument concept is, however, approaching the limit of what it could achieve in terms of $CO_2$ plume observation under favourable conditions, i.e. where the effect of aerosol induced errors are neglected and the SZA is relatively low. A second peak in the average along-track $XCO_2$ enhancements is observed approx. 850 m above (north of) the fourth strongest emitter $E_4$. This enhancement stems from the $CO_2$ plume from the seventh strongest emitter in Indianapolis (labelled as $E_7$ in the top-right panel of Fig. 10) with an annual emission rate of $Q_7 = 0.1\,\mathrm{MtCO_2}\,\mathrm{yr}^{-1}$. Quantifying the $CO_2$ emission rate from such a weak source is, however, not realistic given the low sampling density (especially further downwind) in combination with the weak per-pixel enhancements.

## 6 Conclusions

To follow the progress on reducing anthropogenic $CO_2$ emissions worldwide, independent monitoring systems are of key importance. In this paper, we present the concept of a compact space-borne imaging spectrometer with a high spatial resolution of $50 \times 50\,\mathrm{m}^2$, targeting the monitoring of localized $CO_2$ emissions. We further demonstrate how the instrument concept could resolve $CO_2$ emission plumes from localized point sources like medium-sized power plants, thus having the potential to contribute to the independent large-scale verification of reported $CO_2$ emissions at facility level.

Through radiative transfer simulations using a global trial ensemble, a preliminary, yet realistic, instrument design and an instrument noise model, we show that the expected instrument noise induced $XCO_2$ errors are smaller than 1.1 and 2.0 ppm for 68 % and 95 % of the retrievals, respectively, using the SWIR-2 spectral set-up covering the $CO_2$ absorption bands near 2000 nm. For the SWIR-1 spectral set-up, covering the weaker $CO_2$ absorption bands near 1600 nm, the instrument noise induced $XCO_2$ errors are significantly higher, making it inadequate for the proposed instrument concept. Although the main focus in this paper is on the performance of the proposed $CO_2$ monitoring instrument concept, we could also show that despite the usage of a single spectral window and a relatively coarse spectral resolution of 1.29 nm, scattering by highly complex atmospheric aerosol compositions can be partly accounted for during the $XCO_2$ retrieval on the global scale, limiting the deviation from the true $XCO_2$ to at most 4.0 ppm for two thirds of the retrievals. This gives us confidence that accurate two-dimensional fields of $XCO_2$ enhancements could be retrieved from real spectra measured by the proposed instrument concept. A reasonable a-priori state vector w.r.t. the aerosol properties (e.g. provided through models or a companion aerosol instrument (Hasekamp et al., 2019)) would, however, still be important. As an example, a multi-angle polarimeter instrument is planned to fly together with the $CO_2$ instrument onboard the CO2M mission in order to minimize the systematic $XCO_2$ errors (ESA, 2019).

Using high-resolution $CO_2$ emission data for the city of Indianapolis together with a Gaussian dispersion model, corresponding high-resolution albedo data and additional radiative transfer simulations, we have clearly demonstrated that the instrument is well suited for the task of space-borne $CO_2$ monitoring of large and medium-sized power plants and can (only limited by its own instrument noise) resolve emission plumes from point sources with an emission source strength down to the order of $0.3 \, \mathrm{MtCO_2 \, yr^{-1}}$. This is well below the target emission source strength of $1 \, \mathrm{MtCO_2 \, yr^{-1}}$, hence leaving significant margin for additional error sources and aspects not yet addressed here.

Given the results from this first performance assessment, the proposed instrument concept demonstrates a clear potential for the independent quantification of $CO_2$ emissions from medium-sized power plants ($1$–$10 \, \mathrm{MtCO_2 \, yr^{-1}}$), which are currently not targeted by other planned space-borne $CO_2$ monitoring missions. On the local scale (Indianapolis), we have constrained the present analysis to one day in July using a rather simplistic Gaussian dispersion model that assumes constant atmospheric stability and (unidirectional) horizontal wind speed. It might be that the ability to resolve the $CO_2$ emission plumes becomes more, perhaps even too, challenging under certain more realistic conditions. Nevertheless, these first results are certainly promising and encourage further studies.

The high spatial resolution needed to resolve the emission plumes from localized sources like medium-sized power plant does, however, imply limitations in terms of spatial coverage, arising from the narrow swath (50 km assuming 1000 detector pixels in the spatial dimension) and the forward motion compensation. Hence, a single satellite with the proposed instrument concept could not quantify $CO_2$ emissions at local to regional scale with dense global coverage and high temporal resolution, but would have to be restricted to some pre-defined targets. The relatively compact design with a single spectral window could, however, allow for the deployment of a fleet of instruments and hence independent monitoring of localized $CO_2$ emissions on a larger scale with high temporal resolution. As an alternative to a fleet of satellites, the proposed instrument concept could also prove useful in synergy with a space-borne $CO_2$ lidar (e.g. Kiemle et al., 2017), where the passive spectrometer would

benefit from the lidar's accuracy and knowledge on the light path and the lidar would benefit from the spectrometer's imaging capability.

With the successful demonstration in this paper, i.e. that $CO_2$ emission plumes from medium-sized power plants can be resolved from space with a compact, yet realistic, instrument design, the next step will be to analyse the ability to quantify the corresponding $CO_2$ emission rates from the two-dimensional fields of synthetically retrieved $XCO_2$ enhancements. This follow-up study will be conducted for different seasons (with varying surface albedo and solar zenith angles), meteorological conditions and emission source strengths using large eddy, rather than Gaussian, modelling of the $CO_2$ plume dispersion. Although the effect of aerosols has partly been assessed on the global scale in this study, information on the properties and distribution of aerosols should be included also in the local scale simulations in order to better understand the instrument's ability to resolve and quantify localized $CO_2$ emissions under more realistic conditions. Such an in-depth aerosol analysis is, however, the task of further future studies.

*Data availability.* Hestia Project data at $50 \times 50 \, \mathrm{m}^2$ spatial resolution are available from KG upon request (Hestia project data at original $200 \times 200 \, \mathrm{m}^2$ spatial resolution are available at https://doi.org/10.18434/T4/1503341). Sentinel-2 data are available at https://scihub.copernicus.eu/dhus/#/home. ECOSTRESS Spectral Library data are available at https://speclib.jpl.nasa.gov. CarbonTracker CT2017 data are available at http://carbontracker.noaa.gov. SLUM data are available at http://www.met.reading.ac.uk/micromet/LUMA/SLUM.html.

*Author contributions.* JS developed the instrument noise model, performed the simulations and wrote most of the manuscript. DK lead the instrument design work. JW performed the spectral sizing. CP assisted in developing the instrument noise model and performed the FMC analysis. IS did the optical design. KG and JL developed and provided the Hestia Project $CO_2$ emission data. JS, DK, JW, CP, IS, AR and AB defined the mission concept and instrument design. AR and AB lead the study.

*Competing interests.* The authors declare that they have no conflict of interest.

*Acknowledgements.* We kindly acknowledge all persons and institutions that made their data available to us for this study. CARMA v3.0 $CO_2$ emission data for power plants worldwide were provided by Kevin Ummel. Sentinel-2 data used to derive high-resolution surface reflectance data were provided by ESA trough the Copernicus Open Acess Hub. Spectral reflectance data used to scale the Sentinel-2 surface reflectance data were reproduced from the ECOSTRESS Spectral Library through the courtesy of the Jet Propulsion Laboratory, California Institute of Technology, Pasadena, California, USA. CarbonTracker CT2017 data were provided by NOAA ESRL, Boulder, Colorado, USA. Spectral reflectance data for urban materials (see supplement) were reproduced from the Spectral Library of impervious Urban Materials (SLUM) through the courtesy of the University of Reading, U.K. We also thank Peter Haschberger, Claas Köhler, Günter Lichtenberg, Andreas Baumgartner, Christoph Kiemle, Luca Bugliaro, Julian Kostinek and Andreas Luther for valuable input on the mission concept and instrument design and/or helpful comments on a previous version of this manuscript.

*Financial support.* The study was financially supported by the DLR (Deutsches Zentrum für Luft- und Raumfahrt) project CO2MON.

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
