# Peer review of "Towards space-borne monitoring of localized $CO_2$ emissions: an instrument concept and first performance assessment"

_Atmospheric Measurement Techniques, 2019_

## Referee Comment (RC1) · Anonymous Referee #1 · 7 Feb 2020

Manuscript "Towards space-borne monitoring of localized CO2 emissions: an instrument concept and first performance assessment" of Strandgren et al. highlights the importance of global observations to detect and quantify emissions of localized CO2 emission sources such as coal-fired power plants. They presents a concept for satellites to conduct these observations in the future. They explain the proposed satellite/instrument concept and show global and city-scale assessment results based on simulations. The manuscript is well written and presents interesting new results. I recommend publication in Atmos. Meas. Tech. after the comments listed below have been carefully addressed by the authors.

[Figure]

General comments

Detection of an emission plume is not the same as accurate quantification of emissions and the paper including the abstract must make clear what exactly is meant here. Abstract, line 6 following: Sentences: "... the goal is to reliably estimate the CO2 emissions from localized sources down to a source strength of approx. 1 MtCO2/yr," and "Resolving CO2 plumes also from medium-sized power plants (1-10 MtCO2/yr) is of key importance for independent quantification of CO2 emissions from the coal-fired power plant sector.". What does "to reliably estimate the CO2 emissions from localized sources" mean? Please clarify already in the abstract. Is 1 MtCO2/yr the expected 1-sigma uncertainty / detection limit ? If yes, this would mean that the 1-sigma uncertainties of the medium-sized power plants are in the range 10%-100%. Is this good enough? Or is this just good enough for detection of medium-sized emission sources but not for accurate quantification? In this context: Is it good enough if errors are larger than 4 ppm in 32% of all cases?

Specific comments

Page 4, line3: Sentence "With such a dense spatial sampling, ...". This seems to refer to "spatial resolution" mentioned in the sentence before but resolution is not sampling.

Page 4, line 6: Sentence "Wilzewski et al. (2019) recently demonstrated ..." This statement is too strong as the cited paper is still in review.

Page 5, line 9: Is there a reason why "a local equatorial crossing time at 13:00" has been selected?

Table 1: Please add Adet (detector area) as this is used in several equations. Is the aperture circular so that the aperture area can be computed given the listed diameter? Please add the missing information.

Figure 4 (a): The dotted vertical line is at x=0.1 and the label refers to Albedo=0.1 whereas the x-axis annotation lists Albedo times cos(SZA)/PI. If this is not correct then

please correct this.

Page 9, bottom: Please add a reference for the statement that the SWIR-1 albedo is higher than the SWIR-2 albedo. Is this always the case?

Section 4.2, Fig.9, Fig.10: Is the retrieval using the true CO2 profile? If not: are the reported errors including the smoothing error? Do Figs. 9(b) and 10(top) only show noise or are there also systematic XCO2 biases? If yes, where are the biases coming from? Is the bias correlated with the emission plume (e.g., due to aerosols)? Please show retrieved minus true also for Fig. 10. I would expect to see an aerosol-related XCO2 bias correlated with the emission plume.

Typos etc.

Page 12, line 4: Strange sentence: "For the SWIR-2 set-up it is only retrievals over scenes . . .". Probably "it is" needs to be removed.

Page 12, line 14: Add "nm" after "1.29".

Page 14, line 2: "Which effect that is dominating . . .": delete "that".

Page 14: ". . . the Hestia Project was gridded . . .". Replace by ". . . the Hestia Project data set was gridded . . ." or equivalent.

Various places including References: Check CO2 etc and use subscripts where needed, e.g., for CO2 and CH4.

---

## Referee Comment (RC2) · Luis Guanter (Referee) · 17 Feb 2020

The manuscript by Strandgren et al. describes a new instrument/mission concept for the space-based monitoring of atmospheric XCO2 at a higher spatial sampling (50 m) than currently achieved by other missions. The authors carry out a sensitivity analysis to evaluate the performance of the proposed system for XCO2 mapping in terms of driving error sources such as measurement noise, atmospheric scattering and surface albedo heterogeneity. Their results show that the proposed instrument can resolve emission plumes up to ∼0.3 Mt CO2/yr, which is claimed to be sufficient for the mission to become a useful complement to other existing and planned XCO2 missions

measuring at a coarser spatial sampling.

This is a nice study and manuscript in my opinion. The methodology is sound, the results are clearly presented, the manuscript is well written and the topic fits perfectly into AMT's scope, so I recommend publication. I would suggest the authors to address the following general points in their revision:

1) Plume mapping vs emission quantification - I understand that the quantification of $CO_2$ emissions is the core goal of the proposed system (e.g. "the goal is to reliably estimate the $CO_2$ emissions from localized sources" p1 L7). However, the entire analysis in this manuscript is focused on $XCO_2$ retrieval, without any discussion of the subsequent $CO_2$ flux calculation. Here, I wonder whether the latter drives any observational requirement affecting the instrument/mission configuration. For example, does the $CO_2$ flux estimation interpose any requirement on either revisit or overpass time? On the other hand, the analysis of results in Figs.9-10 is highly based on whether or not $XCO_2$ plumes can be visually detected from the retrieval results. But can those "detected plumes" be used to infer $CO_2$ fluxes within the expected accuracy? I reckon that propagating measurement errors all the way to $CO_2$ fluxes is probably beyond the scope of this study, but some overall discussion of the potential and limitations of the proposed mission/instrument for $CO_2$ emission quantification is certainly missing.

2) Cloud screening - I understand that the retrieval can account for aerosol and cirrus, but I miss a discussion on how optically-thicker clouds would be detected and screened out from the processing. Just avoiding cloudy sites in the mission acquisition plan doesn't seem to be enough. As far as I know, either the $O_2$ A-band or the combination of information from two SWIR channels is used for cloud detection in other $CO_2$ monitoring missions (e.g. OCO-2). What would be the approach here?

3) Spectral albedo variations - the authors discuss the effect of surface albedo on their retrieval using simulations based on Sentinel-2 surface reflectance data, but if I understand correctly a constant reflectance value is assumed for the entire fitting

window. However, I think the impact of different spectral signatures within the fitting window should also be tested. This could be especially relevant for retrievals over urban environments, which are not only characterized by highly heterogeneous surfaces, but also by the presence of artificial materials with strong absorption features in the SWIR. See for example Ayasse et al. (https://doi.org/10.1016/j.rse.2018.06.018) or Cusworth et al. (https://doi.org/10.5194/amt-2019-414) for analysis of the impact of surface reflectance on methane retrievals for 10-nm sampling instruments. It might be the case that the decoupling between CH4 and surface reflectance is less challenging for the much higher spectral sampling of the proposed instrument, but I think a test of this effect would be important nonetheless. The authors could perhaps link their Sentinel-2 background image with the ECOSTRESS spectral library, SPECCHIO (https://specchio.ch/) and/or any other spectral library containing impervious/urban materials (e.g. http://www.met.reading.ac.uk/micromet/LUMA/SLUM.html).

Other minor points:

- p6, L1 SNR already defined (p4, L5)

- Table 1 - specs for swath (1000 across-track pixels?), MTF/PSF and uniformity (smile/keystone) would also be useful

- p9, L1, FMC: does this mean that there is a variation of the view zenith angle from +20° to -20° in the along track direction of the image? how is this handled by the retrieval? Please, comment.

- p10, 3rd paragraph, forward simulation set-up:

* Since CH4 and H2O are included in the retrieval state vector for SWIR-1, shouldn't they be varied in the forward simulations as well?

* Should the surface BRDF be considered in the forward simulations in order to evaluate errors from the Lambertian assumption in the retrieval? Not trivial to implement, but probably relevant esp. In the case of urban environments

- p16, L21 Sen2Core -> Sen2Cor

- p20, L1: "can nevertheless be clearly separated from the background" - OK, but is this still enough for a useful estimation of the emitted flux?

- p21 L1 & L18: references to potential synergies with companion instruments - a discussion of the planned strategy for cloud screening would be useful here

---

## Author Comment (AC1) · 6 Apr 2020

First of all we would like to thank the reviewer for taking the time to read and review our manuscript. The helpful comments certainly helped to improve the manuscript and to clarify what we want to demonstrate in this first paper. The referee comments are listed below along with the corresponding reply from the authors (in italic font style) as well as possible changes in the manuscript (in blue italic font style).

[Figure]

**General comments**

Detection of an emission plume is not the same as accurate quantification of emissions and the paper including the abstract must make clear what exactly is meant here. Abstract, line 6 following: Sentences: "...the goal is to reliably estimate the CO2 emissions from localized sources down to a source strength of approx. 1 MtCO2/yr," and "Resolving CO2 plumes also from medium-sized power plants (1-10 MtCO2/yr) is of key importance for independent quantification of CO2 emissions from the coalfired power plant sector.". What does "to reliably estimate the CO2 emissions from localized sources" mean? Please clarify already in the abstract. Is 1 MtCO2/yr the expected 1-sigma uncertainty / detection limit ? If yes, this would mean that the 1-sigma uncertainties of the medium-sized power plants are in the range 10 %-100 %. Is this good enough? Or is this just good enough for detection of medium-sized emission sources but not for accurate quantification? In this context: Is it good enough if errors are larger than 4 ppm in 32 % of all cases?

*This is a valid point and we agree with the reviewer that we tend to be one step ahead when discussing the goals of the instrument concept in terms of $CO_2$ flux quantification. The long-term goal of the instrument concept is the ability to independently derive $CO_2$ fluxes from point sources with an emission rate down to 1 MtCO$_2$/yr. The goal of the present study is, however, to present an instrument concept and demonstrate that it can resolve/detect $CO_2$ plumes from such point sources at all, assuming a realistic instrument design, and thus has the potential of independent flux quantification. A quantitative evaluation of how accurately the corresponding $CO_2$ fluxes can be determined from such satellite observations under various conditions is the task of a follow-up study currently being prepared. With the results of that study, we will be able to quantify with what expected accuracy $CO_2$ fluxes can be determined and thus better define what "reliably estimate" means. Before that follow-up study, which is too comprehensive to include in the present paper, we refrain from specifying a goal for the $CO_2$ flux estimation accuracy as it would be too speculative. To make*

[Figure]

*this clear we have adapted the corresponding part of the abstract, which now reads: "In this paper, we present the concept and first performance assessment of a compact space-borne imaging spectrometer with a spatial resolution of $50 \times 50\,m^2$ that could contribute to the "monitoring, verification and reporting" (MVR) of $CO_2$ emissions worldwide. $CO_2$ emissions from medium-sized power plants ($1$–$10\,MtCO_2\,yr^{-1}$), currently not targeted by other space-borne missions, represent a significant part of the global $CO_2$ emission budget. In this paper we show that the proposed instrument concept is able to resolve emission plumes from such localized sources as a first step towards corresponding $CO_2$ flux estimates". Also the last part of the abstract was a bit too bold at this early point and has been changed to: "…i.e. well below the target source strength of $1\,MtCO_2\,yr^{-1}$. This leaves a significant margin for additional error sources like scattering particles and complex meteorology and shows the potential for subsequent $CO_2$ flux estimates with the proposed instrument concept."*

*We have further revised the conclusions section accordingly. The first paragraph now reads: "To follow the progress on reducing anthropogenic $CO_2$ emissions worldwide, independent monitoring systems are of key importance. In this paper, we present the concept of a compact space-borne imaging spectrometer with a high spatial resolution of $50 \times 50\,m^2$, targeting the monitoring of localized $CO_2$ emissions. We further demonstrate how the instrument concept could resolve $CO_2$ emission plumes from localized point sources like medium-sized power plants, thus having the potential to contribute to the independent large-scale verification of reported $CO_2$ emissions at facility level.". Similarly, the last paragraphs has been revised and now reads: "Given the results from this first performance assessment, the proposed instrument concept demonstrates a clear potential for the independent quantification of $CO_2$ emissions from medium-sized power plants ($1$–$10\,MtCO_2\,yr^{-1}$), which are currently not targeted by other planned space-borne $CO_2$ monitoring missions. On the local scale (Indianapolis), we have constrained the present analysis to one day in July using a rather simplistic Gaussian dispersion model that assumes constant atmospheric stability and (unidirectional)*

[Figure]

*horizontal wind speed. It might be that the ability to resolve the $CO_2$ emission plumes becomes more, perhaps even too, challenging under certain more realistic conditions. Nevertheless, these first results are certainly promising and encourage further studies."*

*"With the successful demonstration in this paper, i.e. that $CO_2$ emission plumes from medium-sized power plants can be resolved from space with a compact, yet realistic, instrument design, the next step will be to analyse the ability to quantify the corresponding $CO_2$ emission rates from the two-dimensional fields of synthetically retrieved $XCO_2$ enhancements. This follow-up study will be conducted for different seasons (with varying surface albedo and solar zenith angles), meteorological conditions and emission source strengths using large eddy, rather than Gaussian, modelling of the $CO_2$ plume dispersion. Although the effect of aerosols has partly been assessed on the global scale in this study, information on the properties and distribution of aerosols should be included also in the local scale simulations in order to better understand the instrument's ability to resolve and quantify localized $CO_2$ emissions under more realistic conditions. Such an in-depth aerosol analysis is, however, the task of further future studies."*

*Moreover we have rephrased small parts of the manuscript where the aspect of $CO_2$ flux quantification is too pronounced.*

*The 1 MtCO$_2$/yr is the target source strength that we want to be able to determine emission rates for and does not represent the uncertainty of the emission estimates. How accurate the emission estimates will be for such sources will be addressed in the upcoming study, as explained above. To clarify, the $XCO_2$ errors are only larger than 4 ppm in 32 % of the cases when aerosols and cirrus are included. Accordingly these errors also include systematic errors and should not be understood/treated as statistical errors. We do realize, that the chosen percentiles and presentation of these systematic errors in the manuscript might be confusing and make the reader think that the errors are statistical. This has been revised throughout the manuscript.*

**Specific comments**

Page 4, line3: Sentence "With such a dense spatial sampling, . . .". This seems to refer to "spatial resolution" mentioned in the sentence before but resolution is not sampling.

*The term "dense spatial sampling" here refers to the large amount of pixels per unit area. To avoid confusion the sentence has been revised and now reads: "With such a high spatial resolution and large amount of ground pixels per unit area, averaging of . . ."*

Page 4, line 6: Sentence "Wilzewski et al. (2019) recently demonstrated . . ." This statement is too strong as the cited paper is still in review.

*The paper by Wilzewski et al. (2019) has now been accepted and published in AMT (https://doi.org/10.5194/amt-13-731-2020). Thus we keep the formulation as it is.*

Page 5, line 9: Is there a reason why "a local equatorial crossing time at 13:00" has been selected?

*13:00 is chosen in order to have 1) the sun high up in the sky leading to a stronger signal and 2) a relatively well developed boundary layer such that the $CO_2$ plumes can be well dispersed vertically. The following sentence has been added to the manuscript: "This orbit is chosen in order to have a well developed boundary layer at overpass together with good radiometric performance (high SNR)."*

Table 1: Please add Adet (detector area) as this is used in several equations. Is the aperture circular so that the aperture area can be computed given the listed diameter? Please add the missing information.

*This is a good point and since we use pixel area rather than pixel pitch, we have replaced the information about the detector's pixel pitch with the detector's pixel area in Table 1 as well as in the text.*

*The aperture is indeed circular such that the aperture area can be computed using the given diameter. This information has been added in the manuscript.*

Figure 4 (a): The dotted vertical line is at x=0.1 and the label refers to Albedo=0.1 whereas the x-axis annotation lists Albedo times cos(SZA)/PI. If this is not correct then please correct this.

*The dotted vertical line is actually at approx. 0.01 (=0.1·cos(70)/π). The figure and corresponding labels and legends is thus correct as it is.*

Page 9, bottom: Please add a reference for the statement that the SWIR-1 albedo is higher than the SWIR-2 albedo. Is this always the case?

*Although not always the case, it is certainly most often the case. We have added a reference (Fig. 7 in https://doi.org/10.1364/AO.48.003322) where this general pattern is visualized for the global trial ensemble used is this study. Additionally, Fig. 1 below shows surface reflectance/albedo data for SWIR-1 and SWIR-2 (and the difference between the two) inside the Indianapolis domain analysed in this study. The SWIR-2 surface reflectance data are the same data used for the study and the SWIR-1 surface reflectance data are derived from Sentinel-2's band 11 (approx. 1560–1660 nm), which is well aligned with the potential SWIR-1 window assumed in this study. The figure clearly shows that the SWIR-2 reflectance is generally lower than the SWIR-1 reflectance, also at urban scale. In addition to the added reference, the manuscript has also been revised to say that the albedo in SWIR-2 is **generally** lower than in SWIR-1.*

*Furthermore the important aspect of higher solar (ir)radiance in SWIR-1, compared to SWIR-2, was missing as an explanation for the consistently higher SNR for SWIR-1. After some further rearrangements, the related paragraph in the paper now reads: "Figure 4a shows the continuum SNR (calculated with Eqs. (1)–(5)) as a function of the scene brightness for the two prospective spectral set-ups SWIR-1 and SWIR-2. The scene brightness describes the conversion from incident solar irradiance to reflected solar radiance and is calculated as the product of the surface albedo and the cosine of the SZA, divided by $\pi$, hence assuming a Lambertian surface. For the reference scene (albedo $= 0.1$, SZA $= 70$), the continuum SNR is approx. 180 and 100 for SWIR-1 and SWIR-2, respectively. The consistently higher SNR for SWIR-1, compared to SWIR-2, is mainly the result of higher solar radiance (see Fig. 3) as well as generally higher surface albedo (see e.g. Fig. 7 in Butz et al. (2009)) in SWIR-1. Looking at the individual contributions from the different instrument noise sources in Fig. 4b, it is clear that the readout noise and signal shot noise are the major contributors, whereas the noise arising from quantization errors, dark current and thermal background radiation has a small or even negligible contribution in comparison. The signal shot noise is, however, smaller than the dark current, read-out noise and quantization noise inside the $CO_2$ absorption bands, where the signal, and hence the signal shot noise, decreases. Note that all noise terms, except for the signal shot noise $\sigma_{SS}$, are constant."*

Section 4.2, Fig.9, Fig.10: Is the retrieval using the true CO2 profile? If not: are the reported errors including the smoothing error? Do Figs. 9(b) and 10(top) only show noise or are there also systematic XCO2 biases? If yes, where are the biases coming from? Is the bias correlated with the emission plume (e.g., due to aerosols)? Please show retrieved minus true also for Fig. 10. I would expect to see an aerosol-related XCO2 bias correlated with the emission plume.

*Yes, the true $CO_2$ profile is used for the retrieval and no smoothing error is included. Figs. 9 and 10 only show the noise. Systematic biases from e.g. aerosols is not analysed at urban scale in this study, but will be investigated in further studies. Since*

*there is no bias, we see no added value of including further panels in Fig. 10, showing retrieved minus true XCO$_2$.*

**Technical comments**

Page 12, line 4: Strange sentence: "For the SWIR-2 set-up it is only retrievals over scenes . . . ". Probably "it is" needs to be removed.

*Revised. The sentence now reads: "For the SWIR-2 set-up, only retrievals over scenes that are darker than our reference scene (albedo $= 0.1$, SZA $= 70$) are expected to have instrument noise induced errors larger than approx. 2 ppm."*

Page 12, line 14: Add "nm" after "1.29".

*Revised.*

Page 14, line 2: "Which effect that is dominating . . . ": delete "that".

*Revised.*

Page 14: ". . . the Hestia Project was gridded . . . ". Replace by ". . . the Hestia Project data set was gridded . . . " or equivalent.

*Revised.*

Various places including References: Check CO2 etc and use subscripts where needed, e.g., for CO2 and CH4.

*Several instances without proper use of subscript in the reference list have been re-*

*vised. In the main text, however, no such instance could be found apart from "CO2M"*
*and "CO2MON", which should be written without the use of subscript.*

[Figure]

[Figure]

**Fig. 1.** (a) True color RGB for the city of Indianapolis. Corresponding surface reflectance/albedo data in SWIR-1 (b) and SWIR-2 (c) as well as the difference between SWIR-1 and SWIR-2 (d).

[Figure]

---

## Author Comment (AC2) · 6 Apr 2020

First of all we would like to thank the reviewer for taking the time to read and review our manuscript. The comments raised by the reviewer certainly helped to improve the manuscript and to clarify several aspects. The referee comments are listed below along with the corresponding reply from the authors (in italic font style) as well as possible changes in the manuscript (in blue italic font style).

[Figure]

**General comments**

1) Plume mapping vs emission quantification - I understand that the quantification of CO2 emissions is the core goal of the proposed system (e.g. "the goal is to reliably estimate the CO2 emissions from localized sources" p1 L7). However, the entire analysis in this manuscript is focused on XCO2 retrieval, without any discussion of the subsequent CO2 flux calculation. Here, I wonder whether the latter drives any observational requirement affecting the instrument/mission configuration. For example, does the CO2 flux estimation interpose any requirement on either revisit or overpass time? On the other hand, the analysis of results in Figs.9-10 is highly based on whether or not XCO2 plumes can be visually detected from the retrieval results. But can those "detected plumes" be used to infer CO2 fluxes within the expected accuracy? I reckon that propagating measurement errors all the way to CO2 fluxes is probably beyond the scope of this study, but some overall discussion of the potential and limitations of the proposed mission/instrument for CO2 emission quantification is certainly missing.

*The two aspects of plume detection and flux quantification and how they shall be addressed in the present paper is a valid point. As implied above, the long-term goal of the instrument concept is indeed the ability to independently derive $CO_2$ fluxes from point sources with an emission rate down to $1\,MtCO_2/yr$. The goal of the present study is, however, to present an instrument concept and demonstrate that it can resolve/detect $CO_2$ plumes from such point sources at all, assuming a realistic instrument design, and thus has the potential of independent flux quantification. A quantitative evaluation of how accurately the corresponding $CO_2$ fluxes can be determined from such satellite observations under various conditions is the task of a follow-up study currently being prepared. It is correct that this follow-up study is too comprehensive to include in the present paper. To clarify the two aspects and the goal of the present paper, the related part in the abstract has been rewritten and now reads: "In this paper, we present the concept and first performance assessment of a*

*compact space-borne imaging spectrometer with a spatial resolution of $50 \times 50$ m$^2$ that could contribute to the "monitoring, verification and reporting" (MVR) of $CO_2$ emissions worldwide. $CO_2$ emissions from medium-sized power plants (1–10 MtCO$_2$ yr$^{-1}$), currently not targeted by other space-borne missions, represent a significant part of the global $CO_2$ emission budget. In this paper we show that the proposed instrument concept is able to resolve emission plumes from such localized sources as a first step towards corresponding $CO_2$ flux estimates"*

*Nevertheless, the we agree that some overall discussion of the potential and limitations of the proposed instrument concept for $CO_2$ flux quantification could be added. A new paragraph has been added to the conclusions section: "Given the results from this first performance assessment, the proposed instrument concept demonstrates a clear potential for the independent quantification of $CO_2$ emissions from medium-sized power plants (1–10 MtCO$_2$ yr$^{-1}$), which are currently not targeted by other planned space-borne $CO_2$ monitoring missions. On the local scale (Indianapolis), we have constrained the present analysis to one day in July using a rather simplistic Gaussian dispersion model that assumes constant atmospheric stability and (unidirectional) horizontal wind speed. Is might be that the ability to resolve the $CO_2$ emission plumes becomes more, perhaps even too, challenging under certain more realistic conditions. Nevertheless, these first results are certainly promising and encourage further studies." This is followed by the discussion on further limitation in terms of spatial coverage, arising from the high spatial resolution and forward motion compensation.*

*The conclusions section has also in general been revised in order to make clear that the goal of this paper is to demonstrate that the target $CO_2$ plumes can at all be detected, and that the aspect of flux estimation will be addressed in a follow-up study. The first and last paragraphs of the conclusions section now read: "To follow the progress on reducing anthropogenic $CO_2$ emissions worldwide, independent monitoring systems are of key importance. In this paper, we present the concept of*

*a compact space-borne imaging spectrometer with a high spatial spatial resolution of $50 \times 50\,m^2$, targeting the monitoring of localized $CO_2$ emissions. We further demonstrate how the instrument concept could resolve $CO_2$ emission plumes from localized point sources like medium-sized power plants, thus having the potential to contribute to the independent large-scale verification of reported $CO_2$ emissions at facility level. . . . With the successful demonstration in this paper, i.e. that $CO_2$ emission plumes from medium-sized power plants can be resolved from space with a compact, yet realistic, instrument design, the next step will be to analyse the ability to quantify the corresponding $CO_2$ emission rates from the two-dimensional fields of synthetically retrieved $XCO_2$ enhancements. This follow-up study will be conducted for different seasons (with varying surface albedo and solar zenith angles), meteorological conditions and emission source strengths using large eddy, rather than Gaussian, modelling of the $CO_2$ plume dispersion. Although the effect of aerosols has partly been assessed on the global scale in this study, information on the properties and distribution of aerosols should be included also in the local scale simulations in order to better understand the instrument's ability to resolve and quantify localized $CO_2$ emissions under more realistic conditions. Such an in-depth aerosol analysis is, however, the task of further future studies."*

*After the above mentioned follow-up study, when the proposed instrument's abilities in terms of $CO_2$ flux quantification are better understood, observational requirements like revisit, overpass time, reasonable number of satellites etc. can be further analysed and defined. For now, no such observational requirements have been clearly defined.*

2) Cloud screening - I understand that the retrieval can account for aerosol and cirrus, but I miss a discussion on how optically-thicker clouds would be detected and screened out from the processing. Just avoiding cloudy sites in the mission acquisition plan doesn't seem to be enough. As far as I know, either the O2 A-band or the combination of information from two SWIR channels is used for cloud detection in

other CO2 monitoring missions (e.g. OCO-2). What would be the approach here?

*It is correct that we will have to be able to identify and screen scenes with thicker clouds and aerosol layers from the data acquired with a single spectral window. The approach we plan is to retrieve $XCO_2$ independently from the two $CO_2$ absorption bands centred near 2010 nm and 2060 nm, respectively, assuming a non-scattering atmosphere. Given accurate spectroscopic data, any differences in the $XCO_2$ retrieved from the two bands will be due to scattering particles as a result of the different optical depths of the two $CO_2$ bands. Hence, scenes with significant scattering can be identified and screened out. We have added the following piece of text in Sect. 4.2 of the manuscript: "Although layers of aerosol and cirrus can be partly accounted for in the retrieval, scenes with thicker clouds and aerosol layers will have to be identified and filtered out in the data processing chain. Such a cloud filter could exploit the different optical depths of the two $CO_2$ bands in the SWIR-2 window by retrieving $XCO_2$ from the two $CO_2$ bands independently (assuming a non-scattering atmosphere) and filter for discrepancies."*

3) Spectral albedo variations - the authors discuss the effect of surface albedo on their retrieval using simulations based on Sentinel-2 surface reflectance data, but if I understand correctly a constant reflectance value is assumed for the entire fitting window. However, I think the impact of different spectral signatures within the fitting window should also be tested. This could be especially relevant for retrievals over urban environments, which are not only characterized by highly heterogeneous surfaces, but also by the presence of artificial materials with strong absorption features in the SWIR. See for example Ayasse et al. (https://doi.org/10.1016/j.rse.2018.06.018) or Cusworth et al. (https://doi.org/10.5194/amt-2019-414) for analysis of the impact of surface reflectance on methane retrievals for 10-nm sampling instruments. It might be the case that the decoupling between CH4 and surface reflectance is less challenging for the much higher spectral sampling of the proposed instrument, but I think a test

of this effect would be important nonetheless. The authors could perhaps link their Sentinel-2 background image with the ECOSTRESS spectral library, SPECCHIO (https://specchio.ch/) and/or any other spectral library containing impervious/urban materials (e.g. http://www.met.reading.ac.uk/micromet/LUMA/SLUM.html).

*It is correct that we assume a constant reflectance for the entire window. We believe that the decoupling between $CO_2$ and surface reflectance indeed will be less challenging with the higher spectral resolution of 1.29 nm assumed for the spectrometer proposed here. Cusworth et al. (2019) show how the retrieval artefacts due to surface reflectance inhomogeneity decrease when the AMPS sensor is assumed, an atmospheric sensor dedicated for $CH_4$ retrievals with a spectral resolution of 1 nm (i.e. similar to the spectral resolution assumed in this study).*

*Following the reviewer's suggestion, we have analysed the spectral reflectance in the SWIR spectral range in more detail using the SLUM (Spectral Library of Impervious Urban Materials) dataset. This dataset has a spectral resolution of approx. 2.5 nm for the spectral range analysed here. Figure 1 below shows spectral reflectances for various urban materials belonging to different sub-categories like asphalt, stone, cement, metal, granite etc. While significant features in the spectral reflectance are evident between 1800 to 2400 nm for several urban materials, the spectral range of the spectrometer proposed in this study (1982–2092 nm, marked black in the attached figure) exhibit little variability. In many cases the assumption of a constant albedo is valid, and for the other cases the reflectance is sufficiently smooth to be fitted using a second order polynomial during the retrieval. In the 2200–2400 nm spectral region, stronger reflectance features are seen, supporting the conclusion by Ayasse et al. (2018), i.e. that surface reflectance features in the 2200–2400 nm region can cause errors in the $CH_4$ retrieval. Hence, we argue that the challenges in decoupling $CH_4$ and surface reflectance at 2200–2400 nm cannot be directly compared to the ability to decouple $CO_2$ and surface reflectance at 1982–2092 nm, even at the same spectral*

[Figure]

*resolution.*

*That being said, we acknowledge that the albedo heterogeneity at urban scale and at such high spatial resolution will be an important aspect to consider in future studies, especially when scattering by aerosols is considered.*

**Specific and technical comments**

p6, L1 SNR already defined (p4, L5)

*Revised.*

Table 1 - specs for swath (1000 across-track pixels?), MTF/PSF and uniformity (smile/keystone) would also be useful

*We have added information about the assumed 50 km swath width. We do, however, argue that information about MTF/PSF and uniformity (smile/keystone) would be too detailed at this point. This information would be more relevant in future studies, when the preliminary design assumed here has been further consolidated or even realized.*

p9, L1, FMC: does this mean that there is a variation of the view zenith angle from +20 to -20 degrees in the along track direction of the image? how is this handled by the retrieval? Please, comment.

*Our FMC approach means that the satellite will operate in a normal push-broom configuration but the ground speed will be reduced by a factor 5. Thus, considering a whole target tile of approx. 50 km along-track length, the viewing zenith angle (VZA) will be approx. +20 degrees for the first across-track row of ground-pixels in the tile. The VZA will then continuously decrease to 0 degrees at the center of the tile. This*

*is followed by a continuous decrease down to approx. -20 degrees at the end of the tile. Measuring the whole tile takes about 70 seconds. The range of VZA for a single ground-pixel is, however, very small, since each ground-pixel is only observed for 70 ms (5 times longer than without FMC). For the study, each tile consists of 1000 ground-pixels in the along-track direction, meaning that if the VZA ranges over 40 degrees for the entire tile, each ground-pixel will have a VZA range on the order of $40/1000 = 0.04$ degrees (assuming that the FMC was perfectly linear in VZA). The information about the VZA for each ground-pixel is used in the retrieval in order to accurately calculate the corresponding light path. The tile is assembled from all individual ground-pixels after the retrieval. Hence, the range of VZA should not be a problem.*

p10, 3rd paragraph, forward simulation set-up:

- Since CH4 and H2O are included in the retrieval state vector for SWIR-1, shouldn't they be varied in the forward simulations as well?

  *In our global trial ensemble, the abundance of $CO_2$, $CH_4$ and $H_2O$ varies between the scenes. Hence, these greenhouse gas concentrations are all varied in the forward simulations, not only $CO_2$.*

- Should the surface BRDF be considered in the forward simulations in order to evaluate errors from the Lambertian assumption in the retrieval? Not trivial to implement, but probably relevant esp. In the case of urban environments

  *As noted above, retrievals are performed for individual ground-pixels under well-defined viewing geometry given the viewing zenith (VZA) and solar zenith angles (SZA) and the relative azimuth. If there is no scattering in the atmosphere, there should not be any BRDF effect on the retrievals since the retrievals estimate an "albedo" parameter. This "albedo" parameter is just the ground reflectivity for the*

*given combination of SZA and VZA - be it a particular value of a non-Lambertian or a Lambertian BRDF. If there is atmospheric scattering, the BRDF plays a role since the scattered light-beams might hit and exit the surface under different angles than the direct light-beam (SZA, VZA). For most parts of this study, we neglect scattering i.e. BRDF effects are by definition neglected as well. Even for the parts of the study that include scattering, we are in a regime of thin particle loads (AOD(NIR) mostly smaller than 0.5). While there might be a BRDF error contribution, we believe that is small compared to the other scattering induced errors (Fig. 6). But, the reviewer is right that we did not include BRDF effects in our study and the reviewer is also correct that this is not trivial. Since the BRDF effects are not decisive, we propose to postpone such an assessment. For clarification purposes, the following sentence has been added to the section with scattering simulations in the manuscript: "Note that errors arising from the Lambertian albedo assumption (BRDF (Bidirectional Reflectance Distribution Function) effects) are neglected in the scattering simulations."*

p16, L21 Sen2Core → Sen2Cor

*Revised.*

p20, L1: "can nevertheless be clearly separated from the background" - OK, but is this still enough for a useful estimation of the emitted flux?

*See response above regarding revision of abstract and conclusions section in order to clarify the goal of this paper in terms of plume detection vs. flux estimation.*

p21 L1 & L18: references to potential synergies with companion instruments - a discussion of the planned strategy for cloud screening would be useful here

*A new sentence including reference regarding the synergy between $CO_2$ and aerosol instruments has been added to the conclusions section: "As an example, a multi-angle polarimeter instrument is planned to fly together with the $CO_2$ instrument onboard the CO2M mission in order to minimize the systematic $XCO_2$ errors (ESA, 2019[1])"*

*For potential synergies with an active instrument, we are not aware of any suitable reference and we propose to stick to the current reference to a $CO_2$ lidar (Kiemle et al. 2017).*

*A discussion about the planned strategy for cloud screening has been added to Sect. 4.2: "Although layers of aerosol and cirrus can be partly accounted for in the retrieval, scenes with thicker clouds and aerosol layers will have to be identified and filtered out in the data processing chain. Such a cloud filter could exploit the different optical depths of the two $CO_2$ bands in the SWIR-2 window by retrieving $XCO_2$ from the two $CO_2$ bands independently (assuming a non-scattering atmosphere) and filter for discrepancies.".*
* * ** * *
[1]ESA: Copernicus $CO_2$ Monitoring Mission Requirements Document, https://esamultimedia.esa.int/docs/EarthObservation/CO2M_MRD_v2.0_Issued20190927.pdf, EOP-SM/3088/YM-ym, 2019.

[Figure]

Fig. 1. Spectral reflectances for various urban materials belonging to different sub-categories (see plot titles) as provided by the SLUM dataset.

Data credit: Spectral Library of Urban Materials (SLUM)

http://www.met.reading.ac.uk/micromet/LUMA/SLUM.html
https://urban-meteorology-reading.github.io/SLUM
https://urban-meteorology-reading.github.io/other%20files/LUMA_SLUM.pdf

Kotthaus, S, TEL Smith, MJ Wooster, and CSB Grimmond 2014: Derivation of an urban materials spectral library through emittance and reflectance spectroscopy, ISPRS Journal of Photogrammetry and Remote Sensing, 94, 194-212. doi:10.1016/j.isprsjprs.2014.05.00

---

## Author Comment (AC3) · 20 Apr 2020

After the manuscript had been accepted for publication in AMT (but not yet published), a small mistake related to Fig. 6, showing $XCO_2$ retrieval errors as a function of total optical thickness (aerosol optical thickness (AOT) + cirrus optical thickness (COT)), was discovered. When reading the AOT and COT for the respective scenes in order to compute the total optical thickness for the scene and generate the plot, the COT was mistakenly added twice. Hence, the total optical thickness presented in Fig. 6 is consistently higher than the actual optical thickness used for the respective radiative transfer simulations.

[Figure]

The radiative transfer simulations themselves as well as the corresponding XCO2 errors etc. are still correct and valid, it is only that the span of the x-axes in Figs. 6a and 6b is too wide and should range from around 0.0 to 0.6 rather than 0.0 to 1.1. Directly related to this, the sentence at Page 10, Line 23-24 (in the AMTD version of the manuscript) describing the range of AOTs in the global trial ensemble should say *"..., aerosol optical thickness (AOT) ranging from 0 to **1.0** with an average of **0.05** (SWIR-2 window) ..."* instead of *"... aerosol optical thickness (AOT) ranging from 0 to **1.1** with an average of **0.18** (SWIR-2 window) ..."*. Apart from that, no text, figure, statement or conclusion in the manuscript is affected by the mistake.

The mistake will be corrected for the final version of the manuscript and the new corrected version of Fig. 6 can already be seen below.
* * *
**Fig. 1.** Corrected version of Fig. 6 in amt-2019-414 (Strandgren et al., AMTD, 2020)